

# Elevation Change of the Antarctic Ice Sheet: 1985 to 2020
Johan Nilsson[1], Alex S. Gardner[1] and Fernando S. Paolo[1]
[1] Jet Propulsion Laboratory, California Institute of Technology, Pasadena, 91109, United States
*Correspondence to*: Johan Nilsson (johan.nilsson@jpl.nasa.gov)
**Abstract.**
The largest uncertainty in future projections of sea level change comes from the uncertain response of the
Antarctic Ice Sheet to the warming oceans and atmosphere. The ice sheet gains roughly 2000 km$^3$ of ice from
precipitation each year and losses a similar amount through solid ice discharge into the surrounding oceans.
Numerous studies have shown that the ice sheet is currently out of long-term equilibrium, losing mass at an
accelerated rate and increasing sea levels rise. Projections of sea-level change rely on accurate estimates of the
contribution of land ice to the contemporary sea level budget. The longest observational record available to study
the mass balance of the Earth's ice sheets comes from satellite altimeters. This record, however, consists of
multiple satellite missions with different life-spans, inconsistent measurement types (radar and laser) and of
varying quality. To fully utilize these data, measurements from different missions must be cross-calibrated and
integrated into a consistent record of change. Here, we present a novel approach for generating such a record. We
describe in detail the advanced geophysical corrections applied and the processes needed to derive elevation
change estimates. We processed the full archive record of satellite altimetry data, providing a seamless record of
elevation change for the Antarctic Ice Sheet that spans the period 1985 to 2020. The data are produced and
distributed as part of the NASA MEaSUREs ITS_LIVE project (Nilsson et al., 2021).



## 1 Introduction

The single largest uncertainty in multi-centennial projections of sea level change comes from the uncertain response of the Antarctic Ice Sheet to warming oceans and atmosphere (Oppenheimer et al, 2019). Reductions in uncertainty will come primarily from developing our understanding of the ice sheet's response to changes in ocean and atmosphere over the observational record. Given the inaccessibility and size of the ice sheet, satellite observations provide the most comprehensive means to assess ice sheet change. One of the most valued observational records comes from a handful of satellite altimeters that, in combination, provide a near-continuous record of elevation-change from 1992 (McMillan et al., 2014; Schröder et al., 2019; Shepherd et al., 2018, 2019; Zwally et al., 2015, 2021). These observations have provided invaluable insights into how the topography of Antarctica has changed over the past 30 years, revealing rapid thinning of key West Antarctic glaciers (Konrad et al., 2017) that have the potential to thin and retreat irreversibly (Joughin et al., 2014; Rignot et al., 2014). Previous studies of the polar ice sheets that used data from a single satellite mission have been hampered by relatively short records over which to assess change. Records longer than 10 to 20 years are needed to reduce the overall uncertainty in elevation change assessments and to reduce the impact of short-term variability on the climate series (Wouters et al., 2013). Therefore, the creation of long-term records is essential for the separation of short-term variability from long-term change. Such records require piecing together observations from numerous satellite instruments, with unique measurement characteristics and sources of error. Previous studies have tried to overcome these issues by either comparing inter mission rates of elevation change (avoiding merging the records) or merging the records at relatively coarse resolution (>50 km) (Davis, 2000; Khvorostovsky, 2012). More recently, progress has been made to construct synthesized records of ice sheet elevation at higher resolution (Schröder et al., 2019; Shepherd et al., 2019; Wingham et al., 2006). Many issues still remain unsolved, including the proper accounting of radar-penetration, slope induced errors, and resolving time-variable and static topography. In this study, we provide new and modified algorithms to mitigate the impact of these issues on the elevation change record. In support of the "Inter-mission Time Series of Land Ice Velocity and Elevation" (ITS_LIVE), a "NASA Making Earth System Data Records for Use in Research Environments" (MEaSUREs) project, we revisit the processing and cross-calibration of more than 30 years of altimetry measurements over Antarctica to provide a state-of-the-art climate record of ice sheet topographic change. Specifically, we combine data from four conventional pulse-limited radar altimeters (Geosat, ERS-1, ERS-2, and Envisat), a dual antenna radar altimeter capable of operating in both Synthetic Aperture Radar Interferometric mode and pulse-limited mode (CryoSat-2), and a small-footprint waveform (ICESat) and photon counting (ICESat-2) laser altimeters, yielding the most comprehensive record of Antarctic elevation change to date (Figure 1 and Table 1).



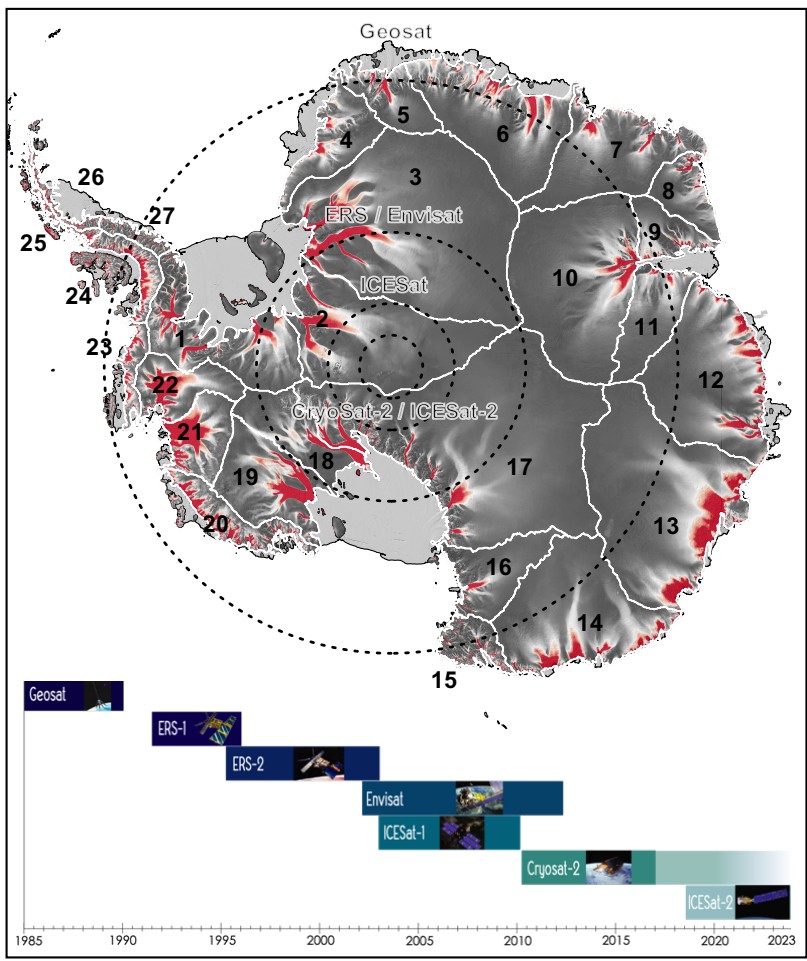

Figure 1: Spatial and temporal coverage of the seven satellite altimetry missions used to produce the elevation change synthesis dating from 1985 to present overlaid on ITS_LIVE velocity map merged with the inSAR phase-based estimates (Mouginot et al., 2019) saturated at 100 m a$^{-1}$ to highlight areas of rapid flow.

## 2 Data

### 2.1 Geosat

The U.S Navy launched the GEOdetic SATellite (Geosat) in March 1985, which operated until September 1989, providing limited Antarctic coverage between ±72° latitude. The main goal of the mission was to provide the U.S Navy with detailed information about the marine gravity field. Geosat operations consisted of two separate missions, where the initial 18 months was the classified "Geodetic Mission" (GM), in a 135-day repeat orbit, ending in September 1986, and the "Exact Repeat Mission" (ERM), in a 17-day repeat orbit, lasting until the end of the mission. The mission carried a Ku-band (13.5 GHz) pulse-limited altimeter providing measurements every





81 670 m along-track (10 Hz), with a pulse-limited diameter of ~3 km. In this study we used "Ice Data Record"

82 (IDR) from the Radar Ice Altimetry Group at NASA Goddard Space Flight Center (GSFC) providing geolocated

83 and corrected surface elevations. Only records with a valid retracking correction and waveforms containing a

84 single return echo were used in the study to reduce noise in the derived surface elevations. We detected the

85 presence of a bias in the Automatic Gain Control (AGC) parameter of 1.23 dB between the Geodetic Mission

86 (GM) and the Exact Repeat Mission (ERM) phases. This is most likely due to the change in orbit and did not

87 affect any of the other parameters, including the surface elevation change.

**2.2 ERS-1 and ERS-2**

89 The European Space Agency (ESA) launched the European Remote Sensing (ERS) satellites in 1991 (ERS-1) and

90 1995 (ERS-2) respectively. They operated continuously between ±81.5° latitude until 1996 and 2003,

91 respectively. Both missions carried conventional pulse-limited Ku-band (13.6 GHz) radar altimeters, with a pulse-

92 limited footprint of ~1.5 km, and an along track resolution of 370 m (20 Hz sampling rate). The two missions

93 operated in a 35-day repeat orbit, though ERS-1 had several shorter mission phases early on that deviated from

94 the standard repeat-track orbit. For this study the "REprocessing of Altimeter Products for ERS (GDR): 1991 to

95 2003" (REAPER), detailed in Brockley et al. (2017) is used to obtain surface elevation measurements. This

96 product contains updated corrections and improved calibrations. For each satellite record we separated the data

97 from the two operational modes, 'ocean' and 'ice', excluding any data used for calibration. The product provides

98 different retracking solutions from which we have chosen to use the ICE1 retracker, otherwise known as the

99 "Offset Center of Gravity" (OCOG) retracker (Wingham et al., 1986) using a 30% threshold of the maximum

100 waveform amplitude. The Ku-chip and the ICE-1 20 Hz quality flag, available in the product, was used to exclude

101 poor quality observations from the analysis.

**2.3 Envisat**

103 The "Environmental Satellite" (Envisat) was launched by ESA in 2002 as a successor to the ERS mission and was

104 officially decommissioned in 2012. Envisat was launched into a 35-day repeat orbit, operating with a pulse-limited

105 radar altimeter with the same footprint, radar frequency, and sample frequency as the earlier ERS missions. For

106 Envisat we used the "RA-2 Geophysical Data Record" (GDR) version 2.1. Only data collected during the period

107 2002 to 2010 were used due to changes in orbit initiated in October of 2010. The GDR product, as with the

108 REAPER product, includes elevations determined using the ICE-1 retracker with a 30% threshold of the

109 maximum waveform amplitude, which we used for this analysis. We applied the same quality filter on the GDR

110 records as with the ERS product, using the Ku-chip and ICE-1 quality flags.

**2.4 ICESat and ICESat-2**

112 The National Aeronautics and Space Administration (NASA) launched the Ice, Cloud, and land Elevation Satellite

113 (ICESat) in 2003, which operated from 2003 to 2009, in a 96-day repeat orbit. The mission carried a novel laser

114 altimeter providing a 70 m beam-limited ground footprint, with 170 m along-track sampling (40 Hz). We used

115 the latest version of the GLAS06 product (release 34), which has been corrected for the "Gaussian-Centroid-

116 Offset" (Borsa et al., 2014), detector saturation and converted to heights above the WGS84 ellipsoid. We did not

117 apply any inter-campaign bias to the ICESat elevations, as there is no consensus that these are required (Borsa et





al., 2019). The records are further edited to remove poor quality observations, using the accompanying quality
flags (elev_use_flh > 0, sat_corr_flg > 2, sigma_att_flg > 0, i_numPk > 0).

The ICESat-2 mission is a follow on mission to ICESat and was launched in October 2018 with the goal of
continuing the long-term altimetry measurements of polar regions (Markus et al., 2017). It carries a new and novel
photon counting laser altimeter that uses 532 nm laser with a pulse repetition rate of 10 kHz and that operates in
a repeat-track configuration over the continental ice sheets. In contrast to its predecessor's single beam, ICESat-
2 collects ground measurements using six individual laser beams arrange in three pairs. Each of the beam pairs is
separated by 3 km and each inter-pair beam by 90 m across track. This configuration allows for a direct estimate
of the across track surface slope that was not directly possible with ICESat's single beam configuration. The beam
limited footprint for each beam is 12 m in diameter sampling every 0.7 m along-track with a repeat frequency of
91 days. In this study surface elevation from the ATL06 product was used following the approach outlined in
Smith et al. (2019, 2020). Here a segmentation filter was used to remove poor quality observations, using a
threshold of 2 m, and further edited using the ATL06 quality flag (*"atl06_quality_summary = 0"*).

**2.5 CryoSat-2**

ESA's CryoSat-2 mission launched in 2010 with the primary purpose of monitoring changes in Earth's Sea and
land ice. This satellite carries a new type of Doppler/delay radar altimeter (Raney, 1998) equipped with a dual
antenna configuration allowing for interferometric measurements of surface elevations. The altimeter system,
referred to as SIRAL, operates in two different modes over the ice sheets; a Synthetic Aperture Radar
Interferometric (SARIn) mode over the marginal areas and a Low-Resolution Mode (LRM) [a conventional Ku-
band pulse-limited radar (identical to ERS and Envisat)] over the ice sheet interiors. The Doppler/delay radar
allows for increased along-track resolution compared to conventional pulse-limited altimetry. The SARIn-mode
has an effective resolution of 350 m along-track and 1500 m across-track. Further, the dual antenna configuration
allows for mapping of the exact position of the surface echo location, by estimation of the across-track look angle
from the difference in path length of the signals between the two antennas. In contrast to previous missions,
CryoSat-2 operates in a drifting orbit, with a 369-day repeat and a 30-day sub-cycle. The drifting orbit offers
improved spatial coverage compared to repeat-track orbits at the expense of larger across track distances. We
processed both the LRM and SARIn modes using the ESA L1b Baseline-C product for the time span 2010-2018
using a custom CryoSat-2 processor described in Nilsson et al. (2016). For the LRM-mode we have chosen to use
a 10% threshold of the maximum waveform amplitude for retracking, similar to Schröder et al., (2019).

**3 Methods**

**3.1 Slope-induced error correction**

The largest source of error in radar altimetry is associated with the effects of surface slope inside the beam-limited
radar footprint. This error stems from an inability to locate the surface from which most of the echo power
originates (off nadir). Because of this, the echo is assigned the location of the sub-satellite point on the Earth
surface. This introduces a slope-dependent measurement error on the order of 0-100 m (Brenner et al., 1983),
which varies with the magnitude of the surface slope. There are a few ways of minimizing the slope-induced error



(Bamber, 1994; Roemer et al., 2007). For this study we used the "relocation method" described in (Nilsson et al.,
2016). This method has been shown to improve surface-elevation retrievals compared to other approaches (e.g.
Schröder et al., 2017). To compute the required surface slope, aspect and curvature, we used the "bedmap2" digital
elevation model from Fretwell et al. (2013) resampled to 2 km horizontal resolution.
**3.2 Elevation change estimation and algorithms**
Surface elevation changes are determined as follows: The local mean topography within a specified search radius
is removed from each mission and mode, leaving only the elevation anomalies that contain the time variable
signal. Artificial trends and seasonal amplitudes in elevation anomalies, that are introduced by changes in surface
scattering characteristics, are reduced proportionally to the correlation with the received radar waveform shape.
Inter-mission biases in seasonal elevation anomalies are further minimized using a normalization scheme that
references all seasonal elevation change amplitudes to those observed by CryoSat-2. A cross-calibration scheme
is applied to adjust and merge elevation change from all missions and modes into a continuous monthly time
series. Lastly, interpolation is used to generate a consistent gridded product with 1920 m horizonal resolution at
monthly time steps from 1985 to 2020. The details of each step are provided in the following sub-sections.
**3.2.1 Removal of time-invariant topography**
To create time series from observations of surface elevations, the time-invariant topography must be removed to
obtain the change signal. This can be done by directly modelling the topography at any given position, e.g., by
fitting a mathematical surface using least-squares, while accounting for the spatial (linear or higher order) and
temporal trends. This rather simple approach, however, has some inherent limitations. When solving for time-
invariant topography one must account for discrepancies between observations originating from: (1) differences
in the orbital geometry of the missions, (2) differences in ascending versus descending range estimates and (3)
differences in measurement density. To account for (1) we employ an iterative prediction-point adjustment to
solve for the topography given a pre-defined grid. For each grid-node, the closest data points inside a specified
search radius are used to compute a new centroid location, when 5 or more data points are available. This centroid
location is used in the next iteration as the new prediction point. This allows us to conveniently follow the
reference orbits (locations of highest data density) to solve for the topography along the satellite ground tracks.
Issue (2) has been handled in different ways (e.g., Flament et al., 2012; McMillan et al., 2014; Moholdt et al.,
2010). We have chosen to solve (2) by separating observations acquired in ascending and descending orbits,
solving for the topography at the same center date, but independently of each other. The differing number of
available observations (3) in each independent solution is handled by allowing for a different number of
coefficients in the mathematical model that is fit to the data. We have provided three different models that can be
used depending on the number of data points available in the local search area. For locations with 15 or more
observations a biquadratic surface (six coefficients) is modeled. When 5 to 14 observations are available a bilinear
surface (three coefficients) is modeled. If there are less than 5 observations the local mean (one coefficient) is
removed and the slopes estimated independently in each direction ($x$ and $y$). A robust least squares approach, M-
estimator (Hubert' T weighting function), is used to solve for the model coefficients (Holland et al., 1977).





Time-invariant surface topography is estimated at each prediction point and removed from the original
observations inside each local search radius. This produces topographic residuals varying only with time. Using
this approach, it is common for the search radius of different along-track centroids to overlap. To ensure that the
best time-invariant topography solution is retained, the correction is only applied if the estimated root-mean-
square (RMS) of the residuals (w.r.t. the time-invariant topography) is lower than the previously computed
solution for the data point in question.
We select different search-radii for the repeat-track (ERS 1/2, Envisat, ICESat, Geosat) and drifting-track
(CryoSat-2) missions. The radius is empirically determined by investigating the residual RMSE from the
algorithm over different types of surfaces. We found that, on a 500 m grid spacing, a search radius of 500 m
provided a good trade-off between the accuracy and computational efficiency of the algorithm for the repeat-track
missions. For CryoSat-2 and Geosat, we found that a higher search radius of 1000 m was needed to provide results
with a comparable RMSE. This larger search radius allows for more ground tracks to be included in the inversion,
reducing the variance of the model residuals. The inclusion of a linear temporal trend in the fit is key to effectively
remove the ascending/descending bias, and to center all data to a common epoch (center date of each mission or
mode).

### 3.2.2 Surface and volume scattering correction

The microwave pulses transmitted by spaceborne radar altimeters at Ku-band frequency (~13.6 GHz) are sensitive
to changes in the dielectric properties of the ice sheet surface (as determined by changes in the snow grain size,
temperature, water content, among others). This effect is aggravated by the variable penetration depth of the radar
signal into the upper layers of the firn-column. Large scale temporal and spatial changes in the scattering horizon
induce changes in measured range, and thus surface elevation, and can introduce long-lived biases in the derived
elevation change rates (Arthern et al., 2001; Davis et al., 2004; Khvorostovsky, 2012; Nilsson et al., 2015;
Wingham et al., 1998). To mitigate this effect, we use a retracking algorithm that tracks the leading edge of the
return waveform (i.e., a maximum amplitude threshold between 10% and 30%). Such retrackers have been shown
to be less sensitive to changes in ice sheet surface properties (Helm et al., 2014; Nilsson et al., 2016; Schröder et
al., 2017). Another key step is removing elevation variability that is correlated with changes in the received radar
waveform shape (Flament and Rémy, 2012; McMillan et al., 2014; Paolo et al., 2016; Simonsen et al., 2017;
Zwally et al., 2005). For this study we approximated the shape of the radar waveform following the definition of
Flament et al., (2012) and Simonsen et al., (2017), using the backscatter (Bs), the leading-edge width (LeW) and
the trailing edge slope (TeS) waveform parameters.
The spatially-variant scattering correction was estimated by computing the local sensitivity gradient (SG) between
each waveform parameter and elevation residuals using a multi-variate least squares inversion. The SG-
parameters were estimated for ascending and descending tracks separately. All waveform parameter time series
were centered and normalized using the mean and standard deviation. Further, parameters were detrended by
applying a difference operator, forming the following least-squares model:



$$\nabla\left(\frac{h - \bar{h}}{\sigma_{dh}}\right) = SG_{Bs} \cdot \nabla\left(\frac{Bs - \overline{Bs}}{\sigma_{Bs}}\right) + SG_{LeW} \cdot \nabla\left(\frac{LeW - \overline{LeW}}{\sigma_{LeW}}\right) + SG_{TeS} \cdot \nabla\left(\frac{TeS - \overline{TeS}}{\sigma_{TeS}}\right) \quad (1)$$


where $\nabla$ is the difference operator, *dh* the elevation residual (elevation relative to time-invariant topography), $\sigma$
the standard deviation and the overbar represents the average value of the parameter.

The SG-parameters were inverted for using the same adaptive search-center approach as described in Sect. 3.2.1.
The estimated SGs were then used to correct each observation within the search cap using the linear combination
of the original waveform parameters and the estimated coefficients. Finally, we apply a linear space-time
interpolation to estimate corrections at locations where the multi-variate fit did not provide a satisfactory solution.

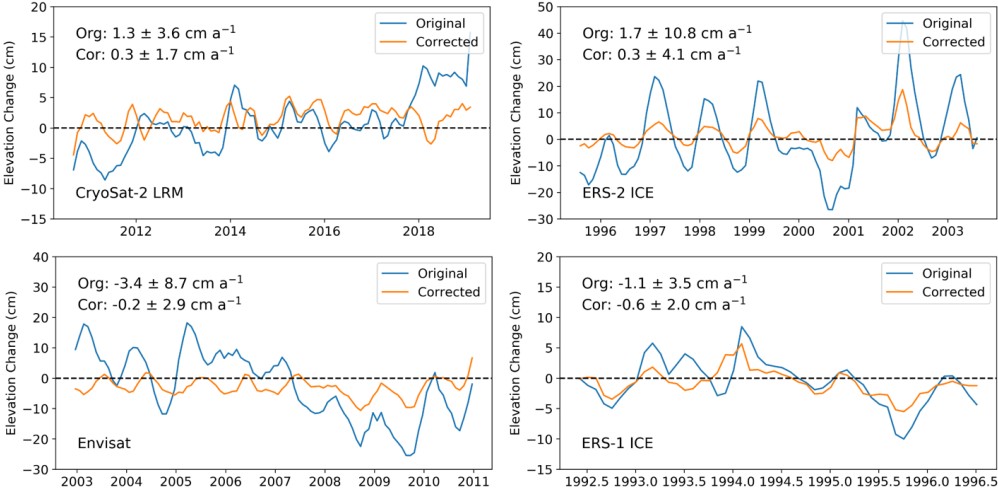


Figure 2. Original and scattering-horizon corrected time series for Lake Vostok in East Antarctica, which for our
purposes is assumed to have a height trend close to zero over recent decades (Richter et al., 2014). A discrepancy
in uncorrected height trends is observed for the various mission due to differences in altimetry processing, orbit
configuration and the quality of the geophysical corrections. Envisat and ERS-2 (Ice) show the largest uncorrected
magnitude in both trend and seasonal signal. Corrected height change records show significantly improved
agreement in trend that are close to zero, given the intrinsic error estimated from the crossover analysis of each
mission.

To determine the optimal search radius for generating the scattering correction, we performed a sensitivity study
over Lake Vostok in East Antarctica (Figure 2). Lake Vostok was selected due to its low surface slope, on average
0.03°, and highly stable surface (Richter et al., 2014), minimizing the impact of the static and time variable
topography in the analysis. After varying the search radius from 1 to 5 km, we found that the 1 km solution
provided the most accurate trend and seasonal amplitude for all missions and modes. We also found that the
absolute magnitude of both the trend and amplitude increased linearly as the search radius increased. We interpret
this result as a decrease in efficiency of the correction, possibly due to de-correlation with increasing
spatial/temporal scales. The use of a 1 km search radius is also computationally efficient as less data are used in
the inversion. Applying these lessons to the ice sheet wide processing, we found that the correction has a minor
impact on the estimated trend for the CryoSat-2 SARIn-mode and the Geosat missions. We also found that the
application of the correction to the SARIn and Geosat data increased the seasonal amplitude of the local (single
grid cell) time series (Figure 3 and 5). Given that there is no physical justification for an increase in seasonal
amplitude, we decided not to apply the correction to the Geosat mission and the SARIn-mode data. For the other
missions, the magnitude of the correction varied across missions and modes of operation, where the largest
changes in trend and amplitude were found for Envisat and ERS-2 ice mode, and the lowest for CryoSat-2 LRM.
By examining the changes in trend and amplitude we found significant spatial patterns, also varying across each
mission and mode, see Figure 3. These patterns show strong correlations to both surface slope/roughness and
signals of metrological origin (Armitage et al., 2014).

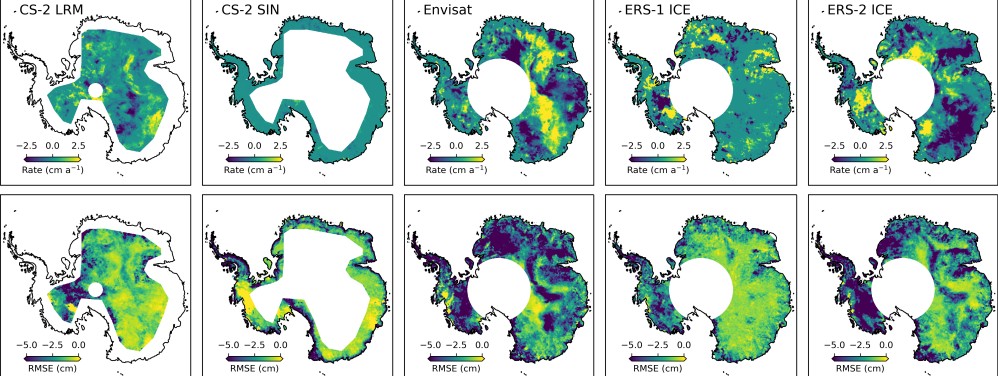


Figure 3: Spatial pattern of the change in rate and RMSE (seasonal amplitude) of the local time series after
correction for temporal changes in scattering horizon (penetration depth). Spatial patterns linked to surface
conditions can be clearly observed. These effects are most prominent for Envisat and ERS-2.

### 3.2.3 Cross-calibration and integration

Removal of the time-invariant surface topography is done internally to each dataset such that elevation residuals
are not aligned to the same surface (see Section 3.2.1). To align elevation anomalies to a common reference we
first solve for inter-mission offsets. These offsets vary regionally (Khvorostovsky, 2012; Wingham et al., 2009;
Zwally et al., 2005), depending on the underlying topography, physical interactions of the radar with the surface,
and differing retracking methodologies. In contrast to previous studies (e.g., Davis, 2005; Khvorostovsky, 2012;
Li et al., 2006; Schröder et al., 2019; Wingham et al., 2006, 2009; Zwally et al., 2005), we estimate these offsets
using a least-squares adjustment. This approach allows for a simple, yet consistent, alignment of multiple relative
elevation anomalies without requiring full overlap between missions to solve. The technique follows the approach
of Bevis etl al. (2014), using the entire multi-mission record to constrain the solution while accounting for trend,
seasonality and inter-mission offsets. The trend is represented by a polynomial, with a maximum order of six; a



four-term Fourier series to account for seasonality; and Heaviside functions to solve for the inter-mission offset
between missions and modes. The design matrix can be written as:

$$h(t) = \sum_{i=1}^{n_p-1} p_i(t - t_r)^{i-1} + \sum_{k=1}^{n_f} s_k \sin\left(\frac{2\pi t}{T_k}\right) + c_k \cos\left(\frac{2\pi t}{T_k}\right) + \sum_{j=1}^{n_j} b_j h \qquad (2)$$


where $n_p$ is the model order, t is the time in decimal years, $t_r$ is the reference time in decimal years ($t_r =$
2013.95), $T_k$ is the seasonal period reference ($T_1 = 1$ and $T_2 = 0.5$), $n_f$ is the number of Fourier series terms ($n_f =$
2) and $n_j$ is the number of missions and modes. To determine the order of the polynomial we use the Bayesian
Information Criterion (BIC: Fabozzi et al., 2014; Schwarz, 1978) to select the polynomial that produces the lowest
BIC-value estimated from monthly binned data.

The cross-calibration is performed on a 2 km polar-stereographic grid (EPSG: 3031) using a variable search radius
of 1-10 km surrounding each grid-cell. The radius is increased until 70% of the time series is filled (monthly) or
the maximum radius is reached. Outliers in the original time series were removed using a 1-year running median
filter where values larger than ten times the median absolute deviation (MAD) were rejected. The model is then
fit to the data using a robust least-squares inversion as in Sect. 3.2.1. Solutions are rejected if the absolute value
of the linear rate is larger than 20 m a$^{-1}$ or if the RMS of the time series relative to the model is larger than 4 m. If
any of the offsets are larger than 100 m the offset is set to zero. The offsets estimated from the least-squares
inversion are then applied to the time series providing an initial cross-calibrated record of elevation change.
Further, the model from the solution is used to filter the time series by omitting observations exceeding ten-times
the MAD of the residuals.

This approach has several advantages; it allows a first order calibration of non-overlapping time series while also
aligning overlapping missions and modes to their common mean. To account for time series that do not fully
conform to our choice of a linear model, a secondary cross-calibration is performed for the four mission-specific
offset coefficients (ERS-1 to ERS-2, ERS-2 to Envisat/ICESat, Envisat/ICESat to CryoSat-2 and CryoSat-2 to
ICESat-2), using the post-fit model residuals. This approach was chosen as it facilitates the estimation of any
residual offsets after removal of the majority of the trend and seasonality, making it simple to estimate the overall
bias between the mission groups. The offsets for groups
ERS-1 to ERS-2, ERS-2 to Envisat/ICESat and CryoSat-2 to ICESat-2 were estimated by taking the median
difference between the two datasets over their respective overlapping time periods. This approach was found to
be suboptimal for the Envisat/ICESat to CryoSat-2 offsets due to the short period of overlap (less than 4 month)
and large changes during the time period 2009-2011. To overcome this limitation, we applied three different
methods, generating five different independent Envisat/ICESat to CryoSat-2 offsets at each search node. Method
1: We fit two second order polynomials to the two residual time series and compute the median offset between
the two functions over a one-year overlap (2010-2011), and the difference between the two intercepts of the
polynomials. Method 2: We applied a Kalman Smoother with a state-space model consisting of a constant local
level and a random-walk trend (Kalman, 1960; Shumway and Stoffer, 1982) that better accommodates the
variability in the time series. The filter was initialized with a variance rate of 1 mm$^2$ a$^{-3}$ (Davis et al., 2012), with
the observational noise given by the RMSE of each residual time series. Initial state-values of the filter were set



to zero for both the level and trend with large initial uncertainties (1e6). The filter parameters were then optimized
using the expectation–maximization (EM) algorithm (Shumway and Stoffer, 1982) with five iterations. The same
approach as in Method 1 was used to generate the two estimates of the offset based on the one-year overlap, and
the differencing of the two intercepts. Method 3: Here the offsets were determined by computing the median
difference between the two missions over the 2010-2011 time period. To determine which of the offsets produces
the best cross-calibration, we apply each offset and compute linear rates of change from 2003 to 2019. These rates
are then compared to rates estimated from unbiased ICESat/ICESat-2 measurements produced by Smith et al.,
(2020), and the offset with the smallest absolute difference was selected. Finally, the selected offsets rate
difference (radar minus laser) is checked against the difference computed without a residual cross-calibration. If
the applied offset did not improve the rate compared to the ICESat/ICESat-2 record, then the residual offset was
set to zero. Following Schröder et al., (2019), we remove outliers in the offsets using a 100x100 km 5-MAD
moving spatial filter. The intermission offsets are then interpolated using a gaussian kernel with a 20 km
correlation length using the nine closest data points. This produces a spatially consistent field of offsets for the
cross-calibration of the elevation residuals. Finally, the individual calibrated elevation residuals for each mission
and mode are averaged to monthly estimates of elevation change for each spatial grid cell, with an associated
standard error. The individual mission/mode time series are then combined and integrated into a continuous record
using the weighted average of the data within each overlapping temporal bin. Weights are specified as the inverse
variance of each mission's accuracy, and the random error estimated from the monthly averaging procedure (see
339 4.1).


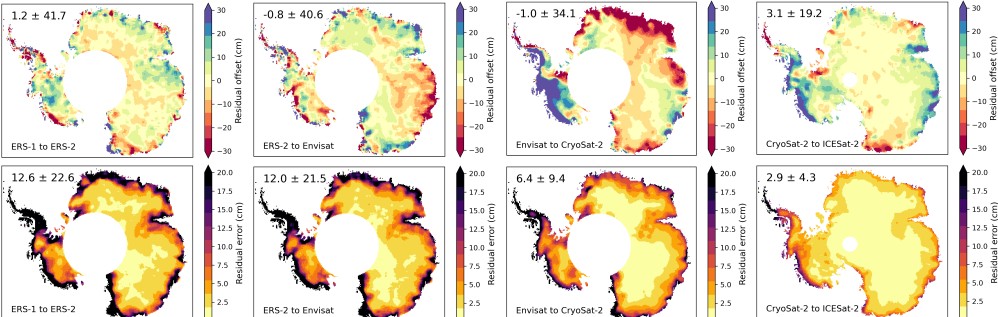

Figure 4. Spatial maps of the residual cross-calibration offset and the corresponding error for the three main inter-
mission transition periods. One should note that here ICESat has been grouped with Envisat in the initial
calibration.

The initial least-squares adjustment provided good alignment between overlapping modes (ocean/ice mode) and
missions (Envisat-ICESat), and a first order correction for the three weakly overlapping missions that allowed
for better estimation of the residual biases from the detrended data. Initial offsets were determined to be as large
as 10-15 m in areas of rapid change such as Pine Island Glacier. However, the least-squares adjustment was
shown to be inadequate when large non-linear elevation changes are present. The magnitude of the estimated
residual cross-calibration error (after least-squares adjustment) (Figure 4) show that most overlapping regions
have a clear correlation with temporal coincident elevation change rates. This pattern is evident in the Envisat to

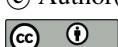



CryoSat-2 transition (Figure 4) for Dronning Maud Land, Wilkes Land, Bellingshausen Sea and the Amundsen
Sea sector (Figure 10: 2010-2012). For the ERS-2 to Envisat transition, we find a clear correlation between the
magnitude of the offsets and the changes in elevation due to variations in surface mass balance in Wilkes's land,
over the 2001-2003 time period (Schröder et al., 2019).

**3.2.4 Normalization of seasonal amplitude**

The radar signals interaction with the surface and sub-surface firn-layers can create artificially large seasonal
amplitudes and trends, as described in Sect. 3.2.2. We corrected for these as best possible using information
contained in the waveform parameters. However, in many cases these corrections are unable to fully correct the
artificial signals introduced by temporal changes in surface and near-surface properties. This behavior can be seen
in Schröder et al., (2019) and in our data, even after the scattering correction has been applied (Figure 5). To
further reduce this effect, we apply an amplitude correction ($h_n$) to each mission to normalize the seasonal signal
over the entire record. We normalized the seasonal amplitudes of the ERS 1 & 2 and Envisat records to match
amplitudes computed from the CryoSat-2. CryoSat-2, which is retracked with a much lower threshold of the
maximum waveform amplitude (10%) for LRM and a maximum gradient threshold for SARin, has been shown
to be less sensitive to changes in surface properties and produces seasonal amplitudes of the same magnitude as
ICESat (Figure 5) (Nilsson et al., 2016). The amplitude normalization was computed for each mission, except for
ICESat and CryoSat-2, after removal of the long-term trend according to:

$$h_n = \left(1 - \frac{a_i}{a_r}\right) \cdot [\alpha_c \, cos(2\pi t) + \alpha_s \, sin(2\pi t)] \qquad (3)$$


where ($a_i$) is the amplitude of the mission ($a^2 = \alpha_c^2 + \alpha_s^2$) and ($a_r$) is the reference amplitude estimated from
CryoSat-2 data. The correction is applied by subtracting it from each individual time series and the normalization
has the effect of producing more homogeneous amplitudes over the entire altimetry record. The application did
not introduce any noticeable shift in the phase of the seasonal signal.

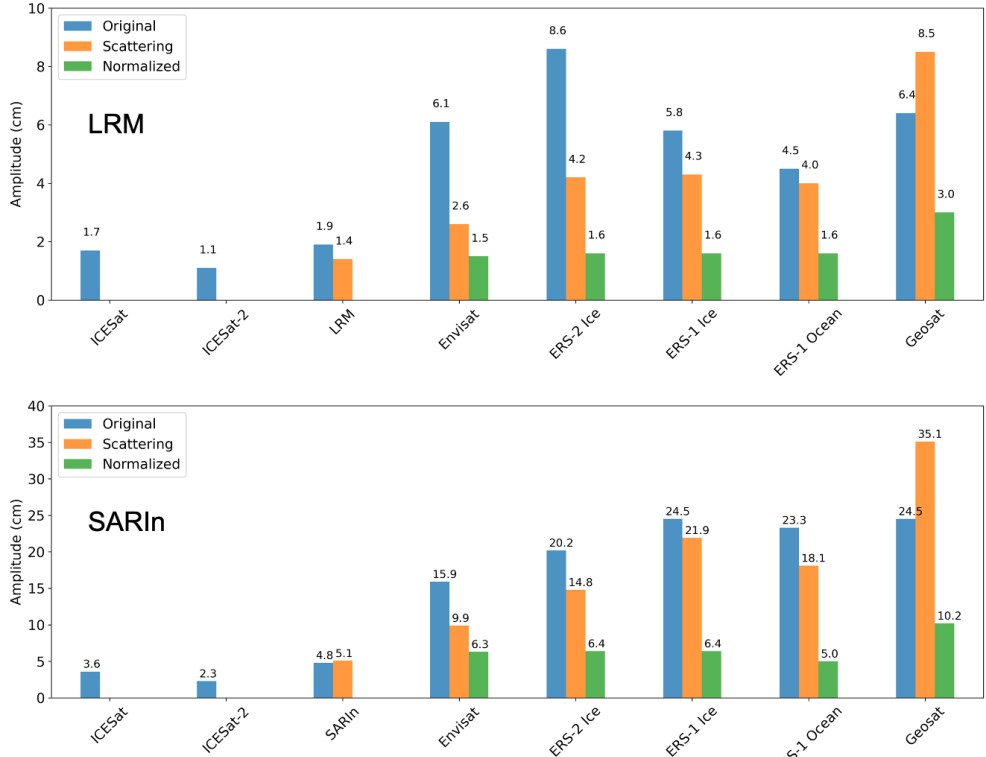


Figure 5. Top: Average seasonal amplitude of the different missions and modes for the CryoSat-2 LRM (top) and
SARin (bottom) mode masks (South of 81.5° S for LRM). The blue bars show the original seasonal amplitude
with no corrections applied, the orange bars show the amplitude once the mission dependent scattering correction
has been applied, and the green bars show the normalized amplitude after adjustment using CryoSat-2 as reference.
For the SARIn-mode and for Geosat we find an increase in seasonal amplitude once the scattering correction was
applied. This is non-physical and thus we have chosen not to apply the scattering correction for these two datasets.
ICESat and the LRM mode show similar magnitude in amplitude and supports the choice of using CryoSat-2 as
reference where the difference is most likely explained by the lower temporal sampling of ICESat. The lower
seasonal amplitude of ICESat-2 is mostly likely due to the short time span used to estimate the amplitude (2-
years).

### 3.2.5 Interpolation, extrapolation and filtering

Collocation (a.k.a. ordinary kriging; Herzfeld, 1992; Nilsson et al., 2016) was used to interpolate the monthly
elevation change estimates onto a 1920 m grid using a maximum search radius of 50 km and a 20 km correlation
length. The 1920 m was chosen to be consistent with the ITS_LIVE grid that accommodates nesting of datasets
at multiple resolutions. An adaptation to Nilsson et al. (2016) is that the local average is replaced by an estimate
from a linear model regressed against both surface elevation (bedmap2) and surface velocity from Gardner et al.
(2018), available at (https://its-live.jpl.nasa.gov), following the approach of Hurkmans et al. (2012) as seen below:



$$m_0 = \beta_0 + \beta_1 h_{DEM} + \beta_2 \log(v) \qquad (4)$$


where ($h_{DEM}$) is elevation values from the DEM and ($v$) are the surface velocity values. The minimum surface
velocity is capped at 50 m per year to avoid introducing noise in the interior parts of the ice sheet and the logarithm
is applied to linearize the range of velocity values.

For the interpolation, the spatial variance is taken to be the mean of the random error estimated from the monthly
averaging procedure. The noise term (diagonal of the error matrix), used in the collocation to weight each
observation, is taken as root-sum-square (RSS) of the variance of the cross-calibration error, mission accuracy
and the random error (see Section 4.1). Further, a minimum error of 5 cm is given to all observations based on
ICESat and ICESat-2 crossover analysis (Section 5.1, Table 2). Prior to the interpolation we remove erroneous
observations using a 100 km radius spatial filter centered at the location of each data value. In this procedure,
following Smith et al. (2020), we remove spatial gradients inside each 100 km cap by fitting a biquadratic surface
and if the observation exceeds a specific threshold it is removed. This threshold is dependent on the local surface
roughness and elevation change rate. If the surface roughness is larger than 60 m and the absolute elevation change
rate is less than 0.2 m a$^{-1}$ (Smith et al. 2020), then the filter threshold is set to 3-MAD otherwise set to 30-MAD
(gross-outliers). This has the effect that the filter is more aggressive in regions of steep topography (Antarctic
Peninsula and the Transantarctic Mountains) while preserving signal in areas of rapid change. In the temporal
domain, and after spatial interpolation, a 12-month median filter is applied to remove outliers exceeding the 10-
MAD threshold. Rejected values in the time series are filled using a gaussian kernel with a correlation length of
3-months.

Differences in satellite orbits cause spatial coverage to vary from 81.5° - 88° S (excluding Geosat that only reached
72° S). The large gap in coverage between the maximum latitude reached and the south pole is referred to as the
pole hole. To create a spatially complete record of elevation change we use extrapolation to fill the pole hole.
Thus, we first average each monthly spatial field to a coarse 20 km resolution, corresponding to the average
correlation length of the elevation anomalies. We then fill the CryoSat-2 and ICESat/-2 pole holes using our
collocation/kriging algorithm (with velocity and elevation terms set to zero), similar to Zwally et al. (2015), using
the 200 closest 20 km averaged values with a correlation length of 100 km and provide each averaged observation
with the aggregated error within each cell. For the 81.5° S missions (ERS 1/2 and Envisat) the unobserved area is
about eighteen times larger than the area for CryoSat-2 and ICESat/-2. This makes common extrapolation
approaches less useful. To overcome this issue, we remove a linear trend and the annual seasonal signal estimated
over the ICESat, CryoSat-2 and ICESat-2 period. The residuals to this model are more homogeneous in the far
field. We extrapolate these residuals to the entirety of the pole hole using the same spatial kriging/collocation
algorithm as previous used but with the velocity and elevation terms set to zero. After the residuals have been
gridded, we add back the model. For both approaches we multiply the predicted errors from the algorithm with a
factor of three to avoid errors that are too small (e.g., less than 5 cm as estimated from ICESat-2). Interpolated
elevation anomalies can easily be included or excluded in any future analysis using the *data_flag* field that is
included with the data product: 0 = no data, 1 = high quality data, 2 = low quality data, 3 = pole hole. The "low
quality data" index is based on surface roughness estimated from an a priori DEM (bedmap2) and is set empirically





starting roughly at the size of the range gate window of the radar altimeters (roughness threshold for Geosat: 30
m, ERS-1/2 and Envisat: 120 m, and CryoSat-2: 240 m).

To estimate basin scale volume changes (Figure 1), we replaced the interpolated values flagged by the surface
roughness criterion with values estimated from a hypsometric relationship (Moholdt et al., 2010; Nilsson et al.,
2015b). Here, the monthly values of elevation change (excluding the values flagged by roughness) were binned
using the median value within 100 m elevation intervals using the hypsometry provided by the DEM (bedmap2).
As in Morris et al. (2020) a linear model was fit to these binned values and used to extrapolate values to areas
flagged as "low quality data". This was done only for the purpose of this paper and is not applied to the final data
product.
**4 Error propagation and validation**
**4.1 Uncertainties of elevation change time series**
An internal crossover analysis was performed to determine the relative accuracy of each mission and mode in a
similar manner as Brenner et al. (2007) and Schröder et al. (2019). We estimated the standard deviation of all
crossovers with a time difference of less than 31-days. Crossovers were binned as a function of surface slope at
intervals of 0.04°. The relative accuracy of each mission or mode was determined from the standard deviation of
the crossovers over low-slope areas (slope < 0.04°). The standard deviation of each individual slope interval is
shown in Fig. 2 and in Table 2. To quantify the spatially varying random error (e.g. driven by topography,
retracking and range corrections) we use the variability inside each monthly interval. To quantify the cross-
calibration error for each time series we use the standard deviation of each offset and add them in quadrature to
estimate the total cross-calibration error, similar to Schröder et al. (2019). We then have the total error ($\sigma_i$) for
each month by summing the individual error sources as:

$$\sigma_i^2 = \sigma_x^2 + \sigma_m^2 + \sigma_c^2 \qquad (5)$$

where ($\sigma_x$) is the mission error derived from the crossover analysis, ($\sigma_m$) the error due to the variability within
each monthly interval and ($\sigma_c$) is the total cross-calibration error.

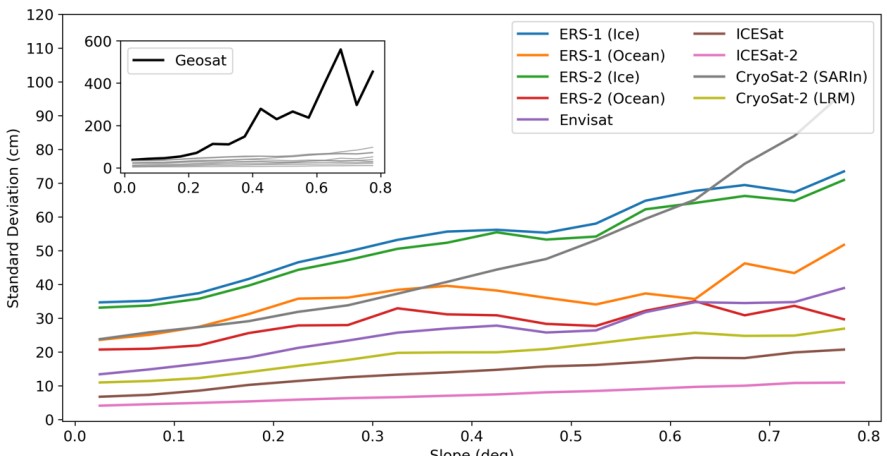

Figure 6. Standard deviation (cm) of intra-mission and intra-mode crossovers for the Antarctic Ice Sheet as a function of surface slope (degrees) for the different missions and modes. Precision decreases quasi-linearly as surface slope increases.

Table 1. Sensor and mode error ($\sigma$) as a function of random ($\sigma_{rand}$) and slope dependent ($\sigma_{slope}$) errors. Slope ($\alpha$) is in degrees. Modelled error ($\sigma$) is based on fitting the following function to the intra-sensor, intra-mode crossover data: $\sigma = \sigma_{rand} + \sigma_{slope}\alpha^2$

| Missions | $\sigma_{rand}$ (cm) | $\sigma_{slope}$ (cm) |
|---|---|---|
| Geosat | 41.6 | 1259.0 |
| ERS-1 (Ice) | 30.0 | 105.2 |
| ERS-1 (Ocean) | 19.4 | 153.0 |
| ERS-2 (Ice) | 29.1 | 86.9 |
| ERS-2 (Ocean) | 17.2 | 105.3 |
| Envisat | 10.8 | 78.5 |
| ICESat | 5.7 | 32.0 |
| ICESat-2 | 3.9 | 8.5 |
| CryoSat-2 (SARIn) | 23.4 | 62.9 |
| CryoSat-2 (LRM) | 9.1 | 41.5 |

## 4.2 Validation of rates of elevation change

To validate the data product, we computed elevation change rates and compared them to rates derived from near-coincident Operation IceBridge (OIB: Krabill et al. (2002)) and pre-OIB data spanning the period 2002 to 2019 using the Airborne Topographic Mapper (ATM: Krabill et al. (2002)) laser altimeter. Elevation change rates for ATM were derived following the approach of Nilsson et al. (2016), where a linear model was solved at each measurement location using a search radius of 175 m. Following the approach of McMillan et al. (2014) and



Wouters et al. (2015), the local slope was used to correct the measurements to the reference track, indicated as
Track_Identifier = 0 in the product. Solutions were rejected if they contained less than two campaigns of ATM
data, the magnitude of linear rate was larger than 10 m a[-1], the standard deviation of the solution exceeded 1 m a[-]
[1], or if the solution contained less than 10 measurements, and if the time span was less than two years. The
elevation accuracy of the ATM sensor family has an estimated error of less than 9 cm (Brunt et al., 2017),
corresponding to an accuracy of roughly 0.5 cm a[-1] over the 18-year measurement period. Operation IceBridge
coverage is concentrated to the western parts of the Antarctic Ice Sheet, providing very limited coverage in the
East. To overcome this limitation, we also use elevation change rates estimated by Smith et al. (2020) that are
based on crossover analyses of satellite laser altimetry (ICESat and ICESat-2: 2003-2019) that has an error of
roughly 10 cm. This corresponds to an error in the rate of elevation change of about 0.6 cm a[-1], which is consistent
with the error observed for ATM.
**4.3. Area integrated error estimation**
Area integrated error for each drainage region, based on the outlines from Zwally et al. (2012), are estimated
roughly following the approach of Nilsson et al. (2016). The total area integrated error is divided into three main
components: the systematic bias, the random error and the rate error estimated in the fitting procedure. These are
then combined in quadrature to produce the total error according to:

$$\sigma_{tot}^2 = \sigma_s^2 + \frac{\sigma_r^2}{n} + \frac{\sigma_{\dot{h}}^2}{n-k} \tag{6}$$


where $\sigma_s$ is the systematic bias, $\sigma_r$ the random error, $\sigma_{\dot{h}}$ the rate error, $n$ is the number of uncorrelated elevation
change estimates (see below) and $k$ is the degrees of freedom in the least squares model ($k = 2$). The systematic
bias and the random error are taken as the average and standard deviation of the difference in rate between the
JPL and ICESat-ICESat-2 (Smith et al. 2020) products for the 2003-2019 period. We compute the error in the
estimated rate using the variance-covariance matrix in the least square fitting procedure according to:

$$\sigma_{\dot{h}}^2 = \bar{\sigma}_m^2 \cdot diag[(X^T X)^{-1}]_{\dot{h}} \tag{7}$$


where $\bar{\sigma}_m$ is the average monthly uncertainty inside the time interval of interest, $X$ is the design matrix of the
linear model, $X^T$ is the transpose of the design matrix, $diag$ the diagonal elements of the array and $()^{-1}$ the
inverse of the dot products. The subscript $\dot{h}$ is the location of the rate error in the diagonal array. To account for
spatial auto correlation $\sigma_r$ and $\sigma_{\dot{h}}$ are divided by $n$. $n$ is estimated by dividing the total area of each drainage
region with the correlation area: $n = A/\pi\rho^2$ where A is the area of the region and $\rho$ is the correlation length. The
errors for each drainage region are summarized in Table 3. The intrinsic quality of each mission was determined
through internal crossover analysis (Section 4.1) of each mode and mission and is summarized in Table 2 and Fig.
6. Analyzing the correlation length of the laser-only versus JPL elevation change differences we found an ice
sheet wide correlation scale on the order of 20-100 km. To be conservative, a correlation length of 100 km was
used to compute $n$.







Table 2. Regionally averaged errors for the synthesized JPL record of elevation change, computed relative to the
unbiased ICESat to ICESat-2 estimate of Smith et al (2020). Errors were determined by differencing 2003-2019
linear rates of elevation change between products. The bias (mean) and error (standard deviation) are computed
for each basin (1-27). AIS, AP, WAIS and EAIS statistics are created using area weighted averages.

| Region | Bias (mm a$^{-1}$) | Error (mm a$^{-1}$) | Area (km$^2$) | Corr. length (km) |
|--------|--------------------|---------------------|---------------|-------------------|
| 1 | -4.1 | 16.0 | 510200 | 112 |
| 2 | -9.1 | 12.6 | 754800 | 62 |
| 3 | -2.3 | 4.7 | 1516300 | 46 |
| 4 | -2.4 | 11.0 | 267300 | 28 |
| 5 | 2.4 | 9.2 | 199700 | 47 |
| 6 | -1.2 | 9.4 | 633900 | 39 |
| 7 | -6.9 | 7.8 | 526000 | 20 |
| 8 | -1.8 | 9.6 | 176900 | 21 |
| 9 | 2.5 | 7.6 | 161100 | 27 |
| 10 | -2.9 | 4.4 | 890600 | 14 |
| 11 | 0.3 | 4.1 | 262300 | 10 |
| 12 | 3.9 | 6.6 | 754700 | 50 |
| 13 | 2.7 | 5.6 | 1142500 | 64 |
| 14 | -1.4 | 5.5 | 742500 | 11 |
| 15 | 6.8 | 27.5 | 150300 | 9 |
| 16 | -2.2 | 6.8 | 269800 | 23 |
| 17 | -2.2 | 5.2 | 1795600 | 59 |
| 18 | 3.5 | 21.3 | 270600 | 29 |
| 19 | 2.3 | 6.3 | 373700 | 30 |
| 20 | 26.4 | 34.6 | 217300 | 20 |
| 21 | 8.9 | 16.3 | 224500 | 51 |
| 22 | 11.0 | 24.4 | 215700 | 71 |
| 23 | -1.0 | 29.1 | 101400 | 13 |
| 24 | -0.4 | 26.7 | 118000 | 14 |
| 25 | -0.3 | 147.8 | 61500 | 13 |
| 26 | -8.3 | 78.2 | 74600 | 8 |
| 27 | 0.1 | 28.1 | 68700 | 12 |
| EAIS | -1.55 | 6.85 | 7653900 | 41 |
| AP | -2.1 | 61.97 | 233300 | 12 |
| WAIS | 5.08 | 18.64 | 1453200 | 57 |
| AIS | -0.55 | 10.08 | 9340400 | 43 |






**5 Results**
**5.1 Accuracy of synthesis**
Previous studies have relied on near co-incident airborne measurements to validate land ice elevation changes
derived from multi-mission synthesis (McMillan et al., 2014; Nilsson et al., 2016; Simonsen and Sørensen, 2017;
Wouters et al., 2015). This approach, however, limits both the spatial and temporal coverage. For Antarctica,
airborne validation data has been collected during austral summer, mostly over rapidly thinning glaciers, such as
Pine Island and Thwaites, in the Western part of the ice sheet, and only with the bulk of the spatial coverage
starting from 2002. The derived errors from these types of local comparisons are then extrapolated to the entire
ice sheet, into regions exhibiting very different surface and metrological conditions. With the launch of ICESat-2
in September 2018 we now have, for the first time, the ability to compare long-term unbiased laser derived rates
of elevation change on a continental scale. For this analysis we compare our synthesized rates of elevation change
to those estimated by (Smith et al., 2020) for the period 2003-2019 for each ice sheet basin (Zwally et al., 2012)
(Figure 1). The results of this analysis are summarized in Table 3. We find an ice sheet wide error of -0.8 ± 7.8
mm a$^{-1}$ (Figure 7c) with a quadratic and linear increase as a function of surface slope in the systematic bias and
random error, respectively (Figure 7d-e). To determine the validity of this comparison we also compared ICESat
/-2 rates with rates from ATM over the time period 2003-2018. Good agreement was found between the two
datasets with an average difference 2.3 ± 22 cm a$^{-1}$ (Figure 7b) over regions with an observed rate of elevation
change from ATM ranging from -15 to 2 m a$^{-1}$. The main discrepancies between the synthesis and the ICESat /-
2 derived elevation change are concentrated over areas of high-relief and over regions with large magnitude of
change, such as Pine Island and Thwaites glaciers. Here, differences larger than 10 cm a$^{-1}$ can be found, and for
the main trunk of Pine Island glacier we find a difference of 2 ± 10 cm a$^{-1}$.

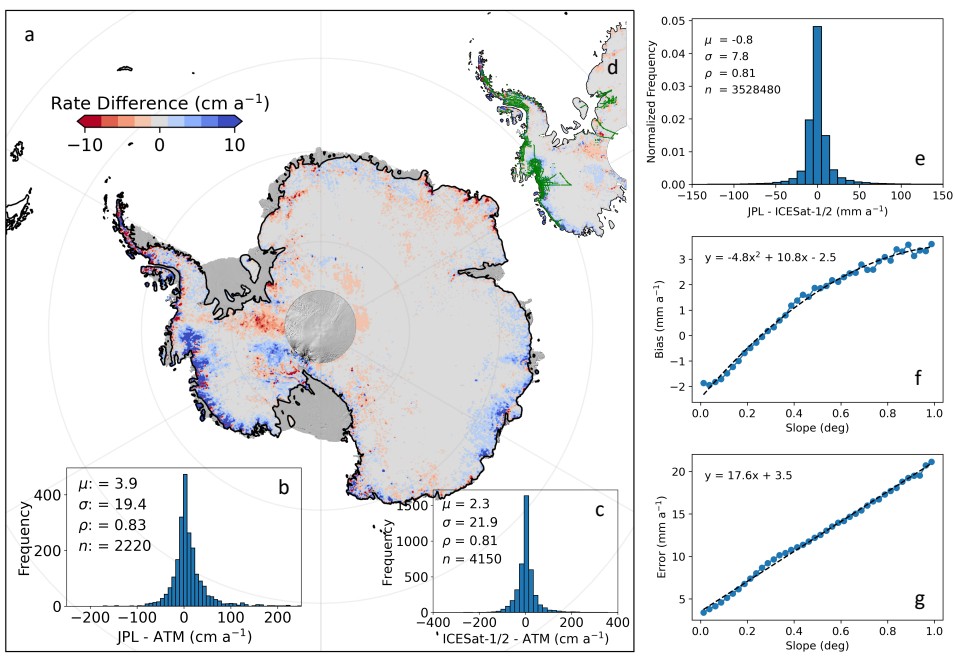




Figure 7. Elevation change validation and comparison using rates derived from ICESat - ICESat-2 and airborne
ATM over the time period of 2003-2019 and 2001-2019, respectively. (a) shows the spatial distribution of the
elevation change differences from this study (JPL) differenced with rates derived from Smith et al. (2020b). (b)
shows the comparison of rates derived from JPL with ATM at locations indicated in (d) by green. (c) shows the
comparison between ICESat - ICESat-2 derived rates with ATM. (e) depicts the ice sheet wide histogram of
elevation change differences and (f-g) the bias (mean) and error (standard deviation) as a function of surface slope.

The relative precision of the different satellite altimeters used in this study range from 5-40 cm over low slope
surfaces (Table 2). Earlier missions such as Geosat, ERS-1 and ERS-2 are roughly three times less accurate than
later missions (Envisat, ICESat/2 and CryoSat-2). However, it was also found that the ERS-1/2 ocean mode was
~30% more precise than ice mode data, bringing it closely in line with the later missions. Unfortunately, the data
coverage of the ocean mode is far lower than the ice mode. For CryoSat-2, the lower relative precision of the
SARIn mode can be attributed to the spatial coverage, with SARIn operating over rougher terrain compared to
the LRM mode that operates over the interior of the ice sheet with a higher along-track resolution (i.e., smaller
footprint). Similar effects were also seen in Schröder et al. (2019). The laser altimetry missions show the lowest
noise levels, on the order of 5 cm over flat areas ranging up to 20 cm for slopes < 0.8°, where ICESat-2 shows a
factor-of-two improvement in precision over its predecessor (ICESat) over all surface slopes.
**5.7 Comparison to other studies and datasets**
Previous long-term Antarctic Ice Sheet elevation change products have been produced by Dresden University of
Technology (Schröder et al., 2019: TUD) and the Centre for Polar Observation & Modelling (Shepherd et al.,
2019: CPOM). These products vary in both resolution and processing methodologies. The TUD product is
provided at a spatial resolution of 10 km and as monthly elevation change estimates. In contrast, the CPOM
product provides elevation change estimates every 5-years at 5 km resolution and basin wide time series of mass
change at quarterly resolution.

The errors reported for our elevation change synthesis are slightly larger than those reported by TUD; this is due
to the difference in retracking and the fitting procedure used to derive the error estimates. Comparing all three
data products to the ATM validation data we find the best agreement with the JPL synthesis. (JPL: $4 \pm 19$ cm a$^{-1}$,
TUD: $6 \pm 20$ and CPOM: $+4 \pm 53$ to $-16 \pm 61$ cm a$^{-1}$). The JPL and TUD estimates where computed from the same
ATM dataset and given the same editing criteria, while values from CPOM are the reported values from Shepherd
et al. (2019). Applying the same analysis to the 2007-2011 and 2011-2016 elevation change solutions provided
by CPOM, we found values of $29 \pm 41$ cm a$^{-1}$ (2007-2011) and $-8 \pm 30$ cm a$^{-1}$ (2011-2016) for the comparison
with ATM, and a weighted average of $-2.2 \pm 33$ cm a$^{-1}$ comparing data from overlapping locations. To further
compare the noise level in the different datasets we use the elevation change from the common 1992-2016 time
period (as CPOM only provides rates in five-year intervals of all products and compare against ICESat-ICESat-2
elevation change rate from 2003-2019. To reduce the impact of difference in time span, we initially compare only
to data between 81.5° and 90° S (pole hole), as this spatial domain only contains ICESat and CryoSat-2
measurements and is thus the most closely aligned in time with the ICESat-ICESat-2 estimate. We also perform
an ice sheet wide analysis, though the time spans are not identical. To compute the noise level, we simply



difference the three rate fields with the ICESat-ICESat-2 derived rates and computed the average and standard
deviation of the differences. This provided the following ice sheet wide results: -0.32 ± 1.70 (JPL), -0.45 ± 1.92
(TUD) and -0.33 ± 2.59 (CPOM) cm a$^{-1}$. For the pole-hole region, 81.5° and 86° S, the following results were
obtained: -0.33 ± 1.17 (JPL), -1.37 ± 1.57 (TUD) and -1.90 ± 3.15 (CPOM).

Comparing the long-term rates for the overlapping time period 1992-2016, we find an overall good agreement for
the three original products. Comparing only values North of 81.5° S, we determine volume change rates of -58, -
48 and -59 km$^3$ a$^{-1}$ for JPL, TUD and CPOM, respectively. Differences are well within the errors for all the three
products. Studying the differences in spatial patterns (Figure 8), using the JPL derived rate as the reference, we
find that the TUD and JPL products agree well over East Antarctica in Basins 10-17 while a larger difference can
be seen in Basin 3. Larger differences between JPL-CPOM compared to JPL-TUD can be observed in East
Antarctica (EAIS). This is likely a result of different methodologies for correcting changes in the radar scattering
horizon within the snowpack. Dividing the estimates into different regions we find for the 1992-2017 time period:
WAIS (JPL: -108, TUD: -100 and CPOM: -106 km$^3$ a$^{-1}$), EAIS (JPL: 61, TUD: 48 and CPOM: 43 km$^3$ a$^{-1}$) and
AP (JPL: -11, TUD: 4 and CPOM: 5 km$^3$ a$^{-1}$). The regional estimates agree well among products where the largest
discrepancy is found to be in the Antarctic Peninsula. Here, both the TUD and CPOM products provide a positive
volume change compared to the JPL-product, highlighting the challenge in obtaining accurate estimates from this
region. Comparing the JPL and TUD products with rates from Smith et al. (2020) (LA) over the time period 2003-
2017 (using the original JPL product with no extrapolation) we find that the two products agree well over WAIS
(JPL: -165, TUD: -164, LA: -200 km$^3$ a$^{-1}$), but biased low compared to LA due to the larger radar footprint. For
EAIS (JPL: 83, TUD: 51, LA: 85 km$^3$ a$^{-1}$) a disagreement of roughly 40% is observed between the TUD and JPL
products, where LA and JPL values are practically identical. In the AP (JPL: -19, TUD: -7, LA: -39 km$^3$ a$^{-1}$) both
products are biased low compared to LA, on the order of 50 - 80% due to limitations in measuring over high relief
topography.

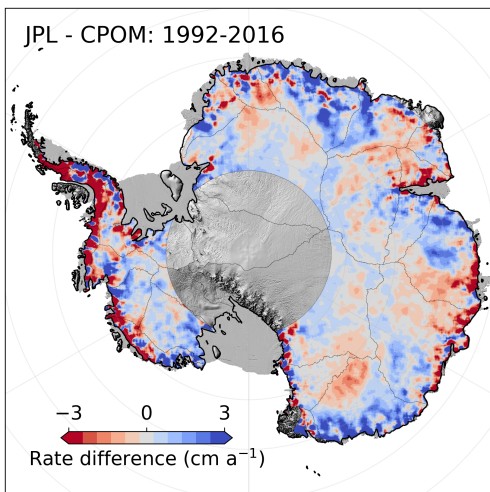 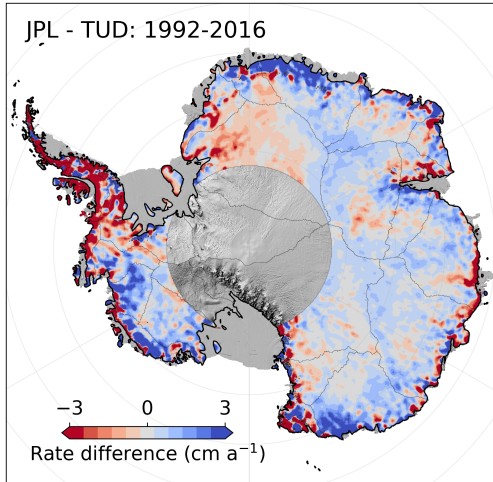





Figure 8: Comparison of overlapping long-term rates from the Technical University of Dresden (TUD) and Center
for Polar Observation and Modelling (CPOM) altimetry product with rates from this study (JPL).

To understand how well these products can capture (and provide insight into) the change/variability of physical
processes of the ice sheets, we compared our result with modeled changes in surface elevations ("*zs*") from the
IMAU firn densification model (FDM: Ligtenberg et al. (2012)) that is forced by 6 hour mass balance components
(snowfall, rain, sublimation and snowmelt), average surface temperature, 10 m windspeed from the Regional
Atmospheric Climate Model, version 2.3p2 (van Wessem et al., 2018). The firn model only simulates changes in
surface elevation due to changes in surface processes and does not account for thinning or thickening resulting
from changes in ice dynamics (flow). To minimize dynamic signals, we mask areas with surface velocities larger
than 30 m a$^{-1}$ using the velocity field provided by the ITS_LIVE project (Gardner et al., 2018) merged with Phase-
Based estimates (Mouginot et al., 2019). The surface elevation long-term trend and acceleration fields (1992-
2016), seen in Fig. 9, show that for Dronning Maud Land and Enderby Land (Basins 4-11) there is generally good
agreement in both the spatial pattern and the sign of the observed and modelled rate of elevation change. For these
regions, the observed change can be attributed to long-term positive accumulation signal (Boening et al., 2012).
However, the magnitude between the modelled and measured rates of change differs by roughly 50%. The
altimetry derived volume change for basins 4-11, over the time period 1992-2016, is estimated at 46 km$^3$a$^{-1}$
compared to a modelled change of 27 km$^3$a$^{-1}$. This disagreement becomes even more prominent for Wilkes Land
(basins 12-14) where the difference between modelled and observed rates of change are larger and of opposite
signs (Figure 9). For these three basins, the estimated difference in volume change is on the order of 36 km$^3$a$^{-1}$
based on the difference in the modelled change of -25 km$^3$ a$^1$ compared to 11 km$^3$ a$^{-1}$ from altimetry. The
magnitude and sign of these results are consistent within all three altimetry products compared to the FDM.
Further, comparing the differences in the magnitude of the seasonal amplitude for 1992-2016, we find that the
TUD product has an annual amplitude that is ~50% larger than the JPL product (5.1 ± 15 versus 2.7 ± 4.9 cm).
Our estimated value of 2.7 ± 4.9 cm compares well with the 2.9 ± 4.1 cm average FDM amplitude for the period
1992-2016. This analysis was not applied to the CPOM product as their provided basin time series are in units of
mass, after a firn correction has been applied.

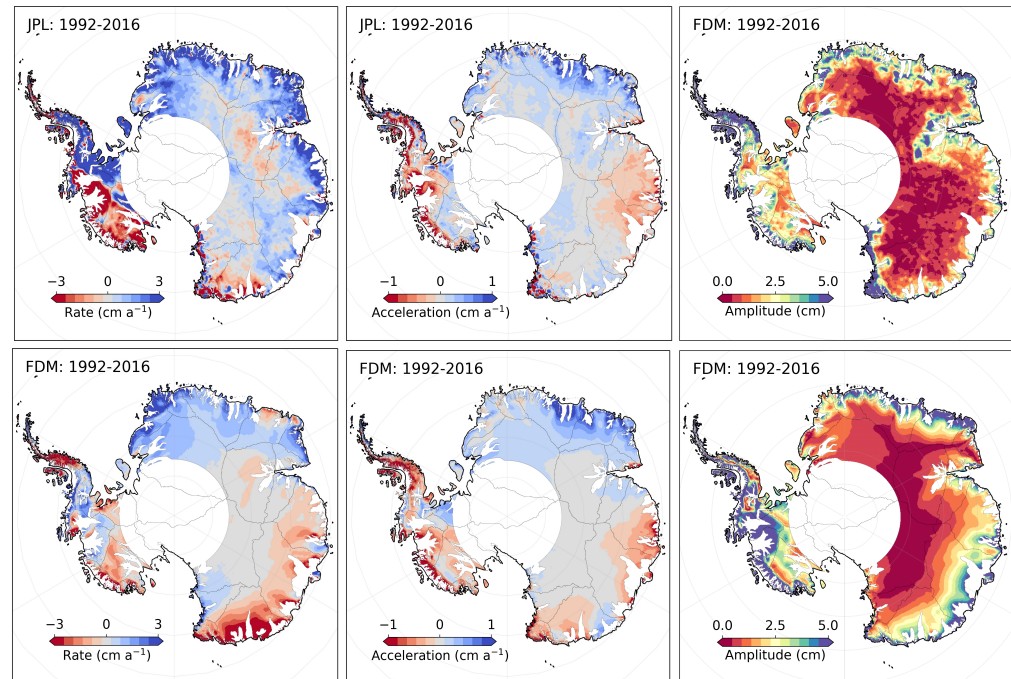


Figure 9: Spatial fields of rates, acceleration and seasonal amplitudes from our product (JPL) and modelled values
from the IMAU firn densification model (FDM). Areas of fast flow (>30 m a⁻¹) have been masked out. The
altimetry data has been smoothed with a 50 km median filter to highlight large scale spatial patterns.

## 5.8 Basin-scale time-evolving volume change


Analyzing the 1992-2020 record of surface elevation (Table 4), including the area between 81.5° and 90° S, we
determine an average rate of volume change of -68 ± 11 km³ a⁻¹ over the entire ice sheet, with large losses from
the West Antarctic Ice sheet (WAIS: -113 ± 6 km³ a⁻¹), and moderate gains for East Antarctic Ice Sheet (EAIS:
+75 ± 5 km³ a⁻¹) that experienced large snow-fall events in 2009 and 2011 (Boening et al., 2012). The most
challenging region to measure elevation change is the Antarctic Peninsula (AP), which has extreme surface relief
and sparse data coverage. We anticipate that any estimate derived from conventional satellite altimetry will be
biased positive due to the inability to measure low elevation signals. That said, we estimate an overall negative
trend for the AP of -27 ± 8 km³ a⁻¹ for the 29-year record (1992-2020) that align closely with other estimates
(Groh el al., 2021; Rignot et al., 2019; Shepherd et al., 2018; Zwally et al., 2021), but is highly dependent on the
applied hypsometric extrapolation (Section 3.2.5). On decadal time scales we find that the large glaciers systems
of Pine Island, Thwaites, Smith and Kohler (Basins 21 and 22) show relatively stable mass loss since the early
parts of the satellite era, with signs of accelerated thinning since 2007-2009. WAIS has seen almost a doubling of
its mass loss in the last decade compared to the two previous decades. EAIS has reverted back to its previous
long-term decadal rate of ~ +8 km³ a⁻¹, in line with the observed 5-year trend from Geosat over Dronning Maud
Land (Figure 12), down from +84 km³a⁻¹ following the anomalous snow-fall during the 2001-2011 period. AP
was in balance and saw little observable change in the first decade (1991-2010), but increased its mass-loss by a





factor of ten in the period of 2001-2011. The mass loss in the last decade was slowed by roughly 50% due to a
positive mass balance anomaly during the period 2016-2018. Over the Geosat time period 1985-1989 a general
stable and small positive rate of 6 ±16 km$^3$ a$^{-1}$was found for the EA1 region (Basins 4-11) (Figure 10 and 12).
This rate remained stable between 1985 and 2009 (~10 km$^3$a$^{-1}$) until the onset of the precipitation event in 2009.
For the EA2 region (Basins 12-15) a shift in both sign and magnitude was observed for the 1985-1989 period.
The mass loss over this period was -54 ± 22 km$^3$a$^{-1}$, and found to be mostly driven by the Totten glacier system
in Basin 13 (Figure 10). This rate is based, however, on heavy extrapolation over the Totten region, due to poor
data coverage for the last two years of the mission, and should be treated with caution.

Table 3. Volume change rates panning1985 to 2020 for Basins 1-27 and aggregate regions. Volume change errors
are computed from the ICESat-ICESat-2 validation procedure, combined with the error in the estimated rate.

| Regions | 1985-1990 | 1992-1994 | 1995-1997 | 1998-2000 | 2001-2003 | 2004-2006 | 2007-2009 | 2010-2012 | 2013-2015 | 2016-2018 | 2019-2020 | 1991-2000 | 2001-2010 | 2011-2020 | 2003-2018 | 1992-2020 |
|---|---|---|---|---|---|---|---|---|---|---|---|---|---|---|---|---|
| 1 | N/A | -20±16 | 34±12 | 34±13 | 36±13 | 51±10 | 16±11 | 25±9 | 24±8 | 30±8 | 20±10 | 14±4 | 23±3 | 20±3 | 19±3 | 20±3 |
| 2 | N/A | -14±17 | 12±11 | -2±11 | 31±11 | 7±7 | 19±7 | -2±6 | 4±5 | 3±5 | 11±5 | -6±4 | 12±3 | 5±3 | 4±3 | 4±3 |
| 3 | N/A | -7±8 | 37±6 | -34±6 | 31±7 | 23±5 | 21±5 | 7±5 | 9±5 | 3±5 | 3±6 | 5±2 | 32±2 | 9±2 | 19±2 | 15±2 |
| 4 | 3±2 | -8±13 | 11±10 | -2±10 | 17±10 | 21±8 | 15±9 | 11±7 | -3±7 | 7±7 | -7±9 | 3±3 | 15±2 | 0±2 | 11±2 | 9±2 |
| 5 | 0±5 | -3±11 | 19±8 | 12±9 | 15±9 | -8±7 | 30±8 | 25±6 | 11±5 | 10±5 | 5±7 | 9±2 | 5±2 | 11±2 | 14±1 | 10±1 |
| 6 | 13±8 | -4±13 | 4±10 | 5±11 | 29±10 | -4±9 | 44±9 | 75±8 | 13±7 | 26±7 | 5±10 | 4±3 | 13±2 | 16±2 | 31±2 | 18±2 |
| 7 | -18±9 | 0±13 | 11±10 | 16±10 | -4±10 | 17±8 | 29±9 | 68±7 | 11±7 | 32±6 | -13±9 | 5±3 | 15±2 | 13±2 | 29±2 | 16±2 |
| 8 | -1±6 | 2±12 | 9±9 | 7±9 | -4±9 | 7±7 | 11±7 | 25±6 | 1±5 | 3±5 | 12±7 | 3±2 | 6±2 | 4±1 | 10±1 | 7±1 |
| 9 | 5±5 | -4±10 | 0±7 | 13±8 | -2±8 | 12±6 | 4±7 | 0±5 | 2±5 | -5±5 | 10±7 | 2±2 | 5±2 | -3±1 | 2±1 | 2±1 |
| 10 | 0±1 | -10±7 | 5±5 | -13±5 | 11±5 | -4±4 | 1±4 | 16±5 | 5±5 | -3±4 | 27±6 | -9±2 | 1±2 | 0±2 | 3±1 | 1±1 |
| 11 | 4±3 | -8±7 | -2±5 | 8±6 | 15±6 | -3±5 | 2±5 | 11±4 | -5±4 | -5±4 | 16±5 | -1±2 | -1±1 | -1±1 | 1±1 | 2±1 |
| 12 | -10±7 | 2±11 | 19±8 | 39±8 | 77±8 | 7±7 | -3±7 | 12±6 | -42±6 | -29±6 | 53±8 | 4±2 | 6±2 | -26±2 | -8±2 | 7±2 |
| 13 | -53±9 | -6±10 | -11±8 | -49±8 | 32±8 | 28±7 | -3±7 | -105±6 | -60±5 | -40±5 | -41±8 | -13±2 | -2±2 | -43±2 | -31±2 | -17±2 |
| 14 | 1±7 | -5±10 | 12±8 | 23±8 | -15±8 | 30±6 | -56±7 | 45±6 | -19±5 | 56±5 | -48±7 | 3±2 | -13±2 | 11±2 | 5±1 | -1±1 |
| 15 | -8±18 | -42±32 | -11±24 | -1±25 | -2±23 | -7±20 | -4±22 | -3±18 | -4±15 | 12±15 | -9±18 | -10±6 | -8±4 | 2±3 | -2±3 | -4±3 |
| 16 | N/A | -12±10 | 9±8 | -3±8 | -4±8 | -3±7 | -1±7 | 5±6 | -4±6 | 3±6 | 0±7 | 4±2 | 1±2 | 2±2 | 3±1 | 1±1 |
| 17 | N/A | -42±19 | 21±12 | -17±11 | 32±11 | -16±8 | -1±9 | 47±8 | -9±7 | 18±7 | 19±8 | 4±3 | -3±2 | 8±2 | 4±2 | 3±2 |
| 18 | N/A | 26±6 | 15±5 | 13±4 | 32±4 | 24±3 | 22±4 | 25±4 | 17±3 | 20±3 | 40±4 | 20±3 | 26±2 | 24±2 | 24±2 | 23±2 |
| 19 | N/A | 4±7 | -11±6 | -34±6 | 3±6 | 11±5 | -6±5 | -11±5 | -5±5 | -4±4 | 9±5 | -12±2 | 1±2 | 0±1 | -4±1 | -5±1 |
| 20 | N/A | -34±26 | -29±20 | -37±21 | -6±21 | -14±18 | -46±19 | -67±13 | -39±11 | -50±11 | 26±16 | -16±6 | -25±5 | -32±4 | -43±4 | -30±4 |
| 21 | N/A | -31±12 | -81±9 | -17±9 | -73±9 | -42±8 | -82±8 | -113±7 | -85±6 | -94±6 | -8±9 | -51±3 | -68±2 | -78±2 | -89±2 | -73±2 |
| 22 | N/A | -8±8 | -31±6 | -13±6 | -28±7 | -13±6 | -57±6 | -90±5 | -68±5 | -66±5 | -2±7 | -20±3 | -32±3 | -58±3 | -62±3 | -43±3 |
| 23 | N/A | -3±20 | -12±15 | 18±15 | -12±15 | 19±13 | 6±14 | -25±9 | -15±8 | -14±8 | 24±12 | -4±4 | 1±3 | -12±2 | -12±2 | -7±2 |
| 24 | -5±7 | -13±27 | 8±21 | 31±22 | 7±21 | 41±18 | -17±19 | -40±13 | -7±10 | -5±10 | 3±17 | 7±5 | 2±4 | -1±3 | -12±2 | -2±2 |
| 25 | -45±23 | 5±23 | 10±17 | -13±18 | -26±18 | -20±17 | -31±18 | -45±14 | -18±13 | 8±12 | -30±16 | -5±8 | -21±7 | -6±7 | -24±7 | -20±7 |
| 26 | 43±18 | -50±23 | -13±17 | -14±18 | -18±17 | -15±16 | 4±17 | -6±12 | -10±11 | 5±11 | -22±14 | -5±6 | -8±5 | -5±5 | -4±4 | -6±4 |
| 27 | 25±9 | -6±22 | -13±16 | 13±17 | -7±16 | 11±15 | -1±16 | -1±11 | 5±9 | -3±9 | -5±12 | 1±4 | 1±3 | 1±2 | 2±2 | 1±2 |
| EAIS | -64±19 | -162±41 | 143±29 | 1±30 | 259±30 | 108±23 | 109±25 | 238±21 | -88±19 | 91±19 | 43±25 | 8±8 | 84±7 | 7±6 | 96±6 | 73±5 |
| AP | 18±36 | -64±42 | -8±31 | 17±33 | -44±32 | 16±29 | -44±31 | -93±22 | -29±19 | 5±19 | -54±26 | -2±11 | -26±10 | -11±9 | -38±8 | -27±8 |
| WAIS | N/A | -66±28 | -116±21 | -36±22 | -48±22 | 36±18 | -147±19 | -255±16 | -170±14 | -178±14 | 109±19 | -68±8 | -74±7 | -135±6 | -166±6 | -113±6 |
| AIS | -46±26 | -292±53 | 19±39 | -18±40 | 167±40 | 160±32 | -82±34 | -110±28 | -288±26 | -82±25 | 98±34 | -62±14 | -16±13 | -140±12 | -107±11 | -68±11 |




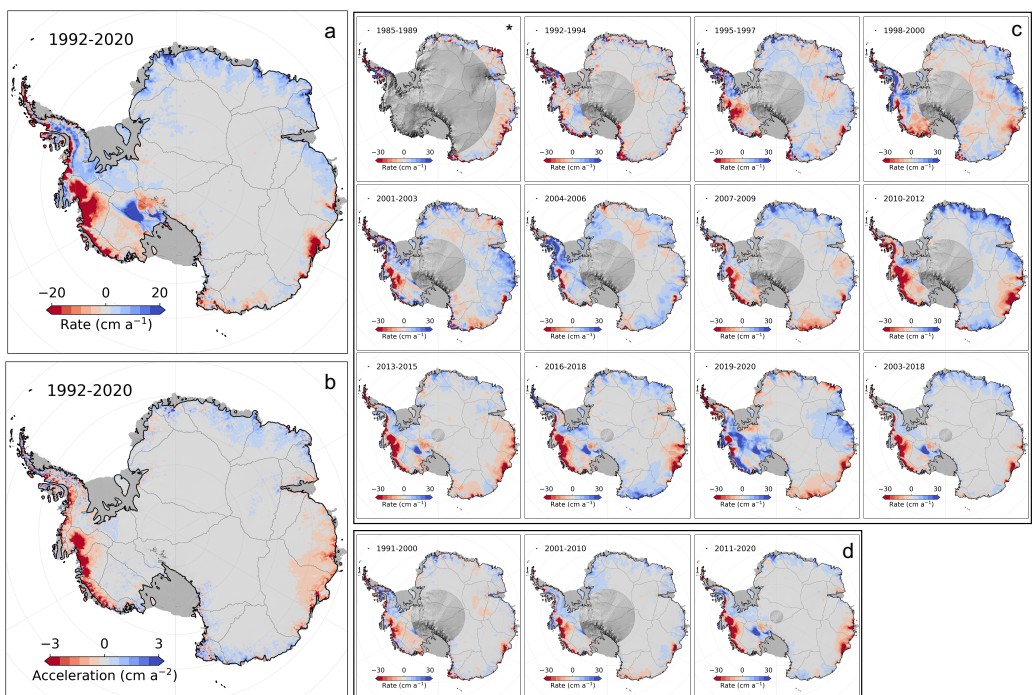

Figure 10. Spatial patterns of Antarctic ice sheet elevation change rates. Long-term elevation change rate (a) and acceleration (b) for the 1992-2020 time period and average rates for (c) 3-year, (d) ICESat – ICESat-2 and (d) 10-year intervals. (*) indicates a five-year interval for Geosat.

Regionally, concentrated rates of thinning from accelerated glacier flow (Gardner et al., 2018; Rignot et al., 2019) are found to spread inland over time due to a regional dynamic imbalance (Shepherd et al., 2019). The marginal areas surrounding the Getz ice shelf (Basin 20) also exhibit negative rates of elevation change but are more localized to the narrow glacier outlets due to inland topographic barriers and time since initiation of thinning. This area saw a large break in the overall long-term trend around 2010 when rapid onset thinning was observed and attributed to short-term variations in both the surface mass balance and ice dynamics (Chuter et al., 2017; Schröder et al., 2019; Gardner et al., 2018). Basin 18, that contains the Kamb Ice Stream, experienced a relatively steady gain in volume over the last three decades resulting from the stagnation of the Kamb Ice Stream some 200 years prior (Catania et al., 2006) (Figure 10). Totten Glacier (Basin 13), part of the EAIS, has been losing mass since the late 1970's (Schröder et al., 2019) with the average trend mostly governed by ice dynamics and short-term variability, and acceleration driven by changes in precipitation (Li et al., 2016). A major change in trend was observed in 2010 when a large-scale thinning of the entire basin could be observed, likely in response to a change in precipitation and possibly changes ice dynamic driven by observed changes in ocean conditions (Khazendar et al., 2013; Li et al., 2016). The activation or reversal in trend of both the Totten and Denman glacier early 2009-2010 has disrupted the long-term equilibrium or gain that has been observed for most parts of Wilkes Land (Basins 12 and 13). A departure from the long-term trend can now be observed for large parts of Wilkes Land in the form of large-scale negative acceleration spreading inland (Figure 10). In Dronning Maud Land and Enderby Land



(Basins 5-8), the previously mentioned snow-fall events in 2009 and 2011 (Boening et al., 2012) are clearly
observed in the regional elevation change trends. This pattern is most prominent along the Weddell Sea coast
where the accumulation signal shows an earlier timing starting already in 2006 (Basins 3 and 4) (Figure 10 and
11). The glaciers flowing into the Bellingshausen Sea have shown a complex pattern of change over the last 29
years. Here, Palmer Land (Basin 24) shows a steady increase in surface elevation over the initial 15 years of the
record, following a long-term positive anomaly in precipitation from 1992. However, a reversal in this pattern
was observed around 2007 where patterns of thinning (McMillan et al., 2014; Schröder et al., 2019; Shepherd et
al., 2019; Wouters et al., 2015) can be observed localized to the major low-elevation outlet glaciers in the regions.
The change can be largely attributed to a change in precipitation amount, with lesser contributions from changes
in ice dynamics resulting from enhanced melting by the ocean (Gardner et al., 2018; Hogg et al., 2017). However,
in the southern part of the Bellingshausen Sea, near Ferrigno glacier in Basin 23, we find a relatively stable trend
during most of the record until 2009 when a large acceleration in ice loss can be observed. This acceleration can
only be partially attributed to changes in ice dynamics (Gardner et al., 2018; Wouters et al., 2015) and it is likely
that changes in precipitation is the major driver of change. Large changes in both spatial and temporal variability
can be observed in the AP region in the last three decades, where large scale reversals of signals can be observed
over different time periods. Here, we find a large-scale positive elevation change anomaly in Basin 23-26,
superimposed on a long-term negative trend, over the time periods 1998-2000, 2004-2006 and 2016-2018. These
changes are linked to changes in the short-term variability of SMB in the region due to increased precipitation.
Examining the rates derived over the ICESat-2 time period (2018-2020) a large positive elevation change signal
can be observed over the WAIS region. This anomaly is directly linked to large scale snow-accumulation, resulting
from an extreme precipitation event in the austral winter in 2019 due to the landfalls of atmospheric rivers
(Adusumilli et al., 2021).

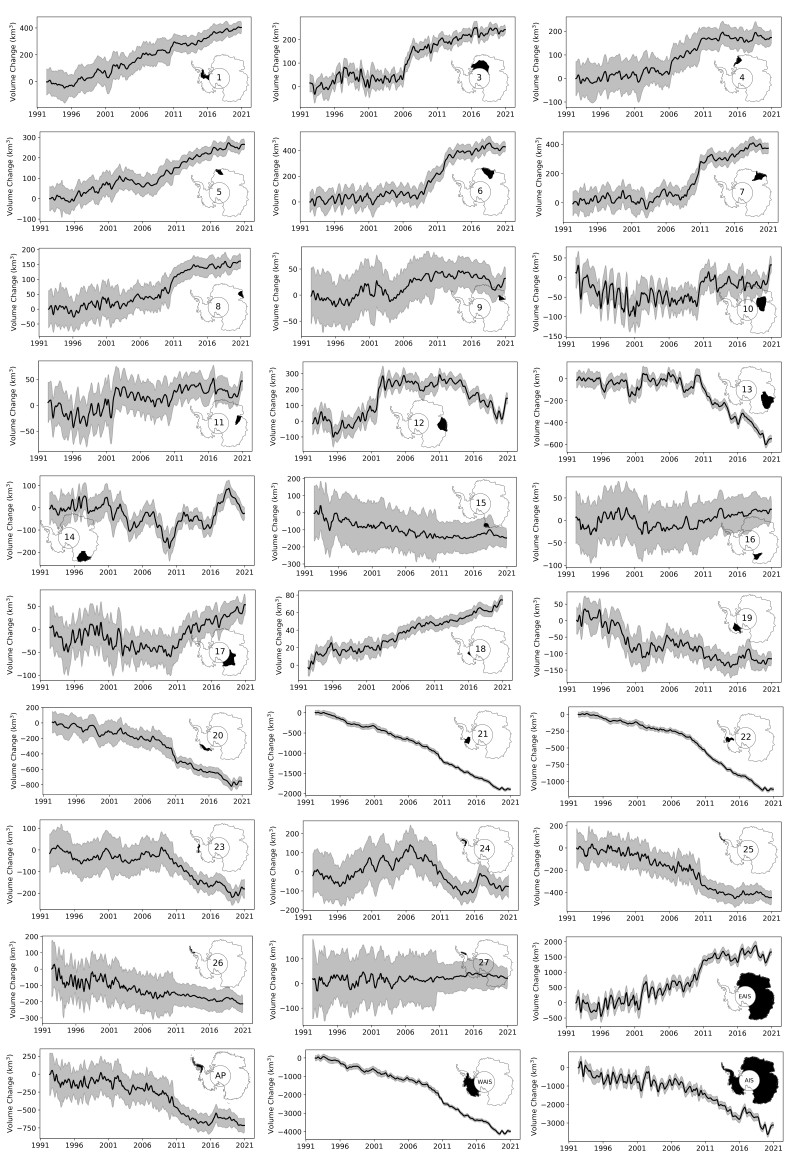

Figure 11. Basin (Zwally et al., 2012) and ice sheet monthly elevation change time series for the period of 1992 to 2020.



## 6 Discussion

We provide a new elevation change product for the Antarctic Ice Sheet that synthesizes over three decades of data from seven different satellite altimeters. To do this we applied slope corrections to all pulse-limited radar altimetry datasets, substantially reducing the overall error in both measured elevation and elevation change rates as can be seen in the crossover quality analysis. Our methodology explicitly separates the time-variable and the static topography in the inversion for elevation change and is one of the major improvements in this study. Removing the time-invariant topography from the time-variable elevation allowed us to more easily accommodate varying spatial scales of correlation inherent to the different processes affecting the altimetry retrievals of elevation. This can easily be conceptualized by noting that correlation lengths are less than <10 km for the time-invariant topography, while elevation change signal are correlated at length scales greater than 50 km in some places. We performed extensive testing over Lake Vostok in East Antarctica and concluded that the optimum search radius for estimating time-invariant topography was 500 m for repeat track missions and 1000 m for drifting-track missions. An extensive investigation was also undertaken to determine the optimum radius for maximizing correlation between the waveform parameters and the time-variable elevation change. From this analysis it was determined that a 1000 m search radius provided the best results in both minimizing the trend and RMS of the residuals. Both spatial and temporal patterns of changes in the scattering horizon (penetration depth) (Figure 2 and 3) of the radar signal further highlights the importance of this correction that can reach magnitudes of several cm a$^{-1}$ (Figure 3). This correction also has a significant impact on the magnitude of the seasonal signal at continent wide scales and can produce reduction of upwards of 50% in the seasonal amplitude of the elevation change signal (Figure 3 and 5).

Cross-calibration of the different missions is likely the most challenging barrier to generating a continuous and accurate record of elevation change. In this study we have taken a somewhat different approach to Schröder et al. (2019) and Shepherd et al. (2019). Here, we work entirely in residual space, after the removal of time-invariant topography. We first apply a least-squares approach to provide an initial inter-mission adjustment. This adjustment is mainly to align overlapping data and modes such as ICESat and Envisat. This approach also has advantages of removing long-term trends and seasonality, allowing us to estimate any remaining offset by examining the residuals to the least-squares model. We find here that the Envisat and CryoSat-2 transition is troublesome, as only a few months of data overlap exist due to the later change in orbit of the Envisat mission and that large ice sheet wide changes occur around this transition. To overcome the sampling problem and the variable elevation change behavior observed for different locations, we investigated several methods to estimate Envisat/CryoSat-2 offsets. Given the availability of high-accuracy ICESat and ICESat-2 elevation change rates we were able to determine which offset provided the most appropriate trend compared to the laser altimetry reference. One should note that we do not use the laser altimetry data to scale or generate the offset, its merely an independent guide to select the most suitable offset produced from the different alignment approaches. This method provides volume changes that are well in line with both the CPOM and TUD products, which provides us with confidence in our approach. Further, it is unfortunate that Envisat changes orbit in late 2010 as it would have allowed almost 2 years of overlap with CryoSat-2. Hopefully this data will be able to be included in the future versions once the issue of how to handle the change in orbit can successfully be addressed, and is work currently being undertaken. As of now, including post orbit change data in the synthesis has the effect of introducing noise

in the Envisat time series and spurious offsets, severely limiting the use of the data. For the Geosat data we include
a caveat for the quality of the cross-calibration. A cross-calibration has been applied but the quality of this
adjustment can vary due to the long-time separation between Geosat (ending in 1990) and the next altimetry
mission (ERS-1: starting in 1992). We recommend that care be taken here and suggest that for regional studies
that a manual post-calibration be applied. The suggestion would be to follow the approach outlined in Sect. 3.2.3
using Eq. (2) varying the degree of the polynomial until satisfactory results are obtained, as seen in Fig. 12.

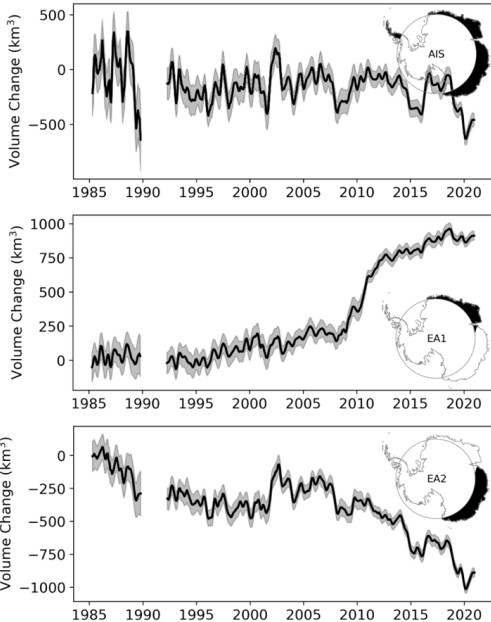


Figure 12. Monthly elevation change time series for the area measured by Geosat (72° S latitude limit) for the
period 1985-2020. The large difference in RMS seen in the Geosat time series for full ice sheet is mostly driven
by observations collected over the Antarctic Peninsula. Regional time series adjustment of Geosat has been
applied to the data align the time series, as suggested in Sect. 6.

Another important correction in the processing is the amplitude normalization, using CryoSat-2 as a reference. It
can clearly be seen in Fig. 5 that even after applying corrections for the change in scattering horizon (e.g.,
penetration bias), the different missions show inconsistent seasonal amplitudes with the older pulse-limited
mission that have seasonal amplitudes that are more than twice that of newer missions (e.g., Envisat, CryoSat-2,
and ICESat/2). This is most likely linked to the higher level of noise in the older sensors (~30 cm vs ~10 cm). The
larger noise levels make it difficult to separate the change in a shifting scattering horizon with time-invariant
topography. Hence, there is need to normalize the different seasonal amplitudes over the different missions, as
there is no physical justification for why they should differ. Here we found that both the ICESat and CryoSat-2
mission showed remarkable good agreement in seasonal amplitude with surface elevation change simulated by
the RACMO firn densification model (Ligtenberg et al., 2012). In the end we selected CryoSat-2 as the reference,
as it provides both higher spatial and temporal sampling compared to ICESat. ICESat-2 was not considered as we
believe that the record currently is too short (only 2 years was used in this study) to provide a viable estimate of



seasonal amplitude. This spatially biases the amplitude to the interior parts of the ice sheet (convergence of the
satellite orbits) where the magnitude of the amplitude is far lower than than closer to the coast. It should be noted
that this correction removes the mean difference in amplitude between missions but does not modulate inter-
annual variability in the amplitude within a single mission.

Large data gaps exist at latitudes exceeding the maximum orbital coverage, this gap is referred to as the pole hole.
In our product we fill the pole hole to provide a spatially complete field to aid in the estimation of ice sheet wide
mass balance and to make the data more usable for modeling efforts. However, we do recognize that our chosen
interpolation method may not be appropriate for regions such as AP and Basins 15-17, which are comprised of
highly variable topography. Therefore, we provide a mask layer (*data_flag*) that identifies *high quality*, *low*
*quality* (high topographic relief), and *pole hole* data. After some investigation we found that applying the
hypsometry method to extrapolating monthly estimates of elevation change produced an improved estimate of
basin scale volume changes when compared to the ICESat-ICESat-2 product. This methodology is not applied to
distributed product. We leave it up to the user of the product to apply their own methodology for extrapolation,
but we recommend that the hypsometric method when generating basin scale mass balance estimates.

Elevation change rates near the pole hole are relatively small, due to low precipitation amounts (Wingham et al.,
2006) and few dynamically active glaciers. Changes in mass within the pole hole only amount to few tens of
gigatons of change (Shepherd et al., 2019), once corrected for firn-air-content. Hence, the interpolation of data to
fill the pole hole only contributes a small part of the overall volume change. In our estimate the overall volume
change is estimated to be 26 $km^3 a^{-1}$ North of 81.5° S over the full 2003-2018 time period using the least-squares
adjustment method and 34 $km^3 a^{-1}$ when adding the residual cross-calibration. This aligns well with the value
estimated from the ICESat-ICESat-2 product of 37 $km^3 a^{-1}$ for the area 81.5°-86° S over the period 2003-2019.
Studying the other two publicly available altimetry synthesis we find that their pole-hole volume estimates are
biased in the negative direction and can be quite large: -65 (CPOM) and -12 (TUD) $km^3 a^{-1}$. This indicates that
using either a constant offset or mission-only derived trends for cross-calibration might not be sufficient for these
areas, as a small error can a have a large impact when integrated over a large region. This further points to the
effectives of using the least-squares adjustment for cross-calibrating non-overlapping records.

Previous altimetry studies of Antarctic mass balance have relied heavily on airborne laser altimetry to provide
validation and estimates of the overall volume change uncertainty (McMillan et al., 2014; Wouters et al., 2015).
However, airborne data are both limited in spatial and temporal coverage, making it extremely difficult to estimate
volume change uncertainties on continental scales. We, for the first time, have used long-term (16-years) unbiased
laser altimetry derived rates of elevation change from Smith et al. (2020) to produce ice sheet wide uncertainties
for our product. This is especially important for East Antarctica where very little validation data exists from either
in-situ or airborne campaigns. Though the rates here are on the order of cm per year, they occur over massive
spatial scales and contribution significantly to the overall ice sheet volume change. 16-years of high-accuracy
laser data allows us to validate these cm trends as the measurement error reduces as a function of time. This dataset
allows us to quantify and validate changes at the mm $a^{-1}$ level, which was previously not possible in East
Antarctica. The overall uncertainty estimates of -0.8 ± 7.8 mm $a^{-1}$ is heavily dominated by the small difference in



the interior areas of the ice sheet but rapidly increase closer to the coast with errors reaching 25 mm a$^{-1}$. In general,
the analysis shows that radar altimeters underperform, relative to laser altimeters, in areas of steep topography
where change signals are largest. Further, we observe that in East Antarctica, the radar record in many places
produces small negative rates, compared to slightly positive rates from laser, indicating residual issues with time-
variable radar penetration biases. These issues are of course not unknown to the scientific community (Arthern et
al., 2001; Davis, 1993; Lacroix et al., 2009; Legresy and Remy, 1997; Nilsson et al., 2015a) and is an area of
active research. However, with this new laser altimetry dataset we now have at least the possibility of quantifying
this type of uncertainty across nearly the entirety of the ice sheet.

Comparing the estimate from this study with the TUD (Schröder et al. 2019) and CPOM (Shepherd et al., 2019)
products we find good agreement over the 1992-2016 time period, with difference within the error budgets of the
respective products. This agreement is a good indicator that all three products provide consistent results given the
different processing methodologies for areas below 81.5° S. Analyzing further, we find that the main difference
between products is in the overall noise levels. Given the different comparisons we find that, on average, our
product has lower noise and agrees most closely with the laser-altimetry validation data. We attribute this
improvement in noise characteristics to the improved processing techniques.

Another, important improvement is the normalization of the seasonal signal across mission. Though this
correction is not perfect, it has lowered the magnitude of the average seasonal signal to a level comparable to the
simulated values of elevation change from the RACMO FDM product (Ligtenberg et al., 2012). Accurate
quantification of the "seasonal breathing" of the Antarctic ice sheet is important component to estimated rates of
snowfall. However, we do find discrepancy between the altimetric and modelled rates of change for East
Antarctica, with rates of change differing in places by 200% to 300%. We further find that the direction of change
can have opposite sign between modeled and observed rates, as can be seen in the Wilkes Land region. This
indicates that the current generation of firn densification models, though highly successful in representing the
main components governing ice sheet mass balance, still cannot fully capture all the complex interactions driving
changes in surface elevation. This of course has large implications for estimating the East Antarctica mass balance
as the correction for firn-air-content can be as large as 100% of the measured altimetry signal in some basins
(Smith et al., 2020). However, several new firn models are expected to become available within the near future,
which will greatly help the community to quantify both the error in these models and to help improve our
understanding of the processes driving the ice sheet mass balance.
**7 Data and code availability**
Data can be found and downloaded from (http://its-live-
data.jpl.nasa.gov.s3.amazonaws.com/height_change/Antarctica/Grounded/ANT_G1920V01_GroundedIceHeigh
t.nc) [Nilsson et al., 2021; DOI registration in progress]. The code and algorithm used to generate the product
are part of the captoolkit – Cryosphere Altimetry Processing Toolkit and can be found
(https://github.com/fspaolo/captoolkit).



**8 Summary and conclusion**
In this study we have provided a 36-year record (1985-2020) of elevation change for the Antarctic ice sheet
derived from seven altimetry missions combining both laser and radar measurements. Elevation changes were
derived from measurements of surface elevation by first removing the time-invariant topography for each mission
and applying corrections for varying surface penetration depth to radar altimetry data. The different sensors and
modes where cross-calibrated and merged into a continuous record of elevation change, using a combination of
interpolation and extrapolation techniques to construct a consistent spatio-temporal dataset for the scientific
community.

Between 1992 and the later parts of 2000's, the Antarctic ice sheet was in near balance, with modest EAIS gains
equaling WAIS losses. In the later parts of the 2000's accelerated WAIS losses outpaced EAIS gains, leading to
significant net decrease in ice sheet volume. This accelerated loss has been attributed to increased ocean melting
and changes in precipitation (Shepherd et al., 2018) East Antarctica has also seen changes over the last 30 years,
where large swaths of Wilkes Land are now showing accelerating negative elevation change starting around the
year 2010 and likely stemming from changes in precipitation/firn, and possibly ice dynamics from the Denman
and Totten glacier systems. The Dronning Maud Land region has started to show extensive elevation gain due
significant increases in snowfall beginning circa-2009. However, one of the main questions still remains: is EAIS
losing or gaining mass? With these long-term improved datasets, in combination with accurate firn-modelling,
we may soon be able to answer this question. The western parts of Antarctica have seen both consistent and
accelerated mass loss over the entire altimetry record dominated by the glacier systems of Pine Island and
Thwaites. These areas now show drawdowns for hundreds of kilometers inland, and currently show no signs of
slowing down. The Antarctic Peninsula also shows signals of major mass loss, but the long-term accuracy of those
estimates is hard to quantify due to inherent limitations of radar measurements over these types of rugged terrain.
We can, however, say with confidence that large changes due to a complex mix of atmosphere and ocean forcing
have accelerated mass loss in the Bellingshausen Sea over the length of the record (Gardner et al., 2018; Hogg et
al., 2017; Wouters et al., 2015). This region was relatively stable for two decades but started to show a large
change in behavior from its original trend in the 2008-2010 period.

It is our hope that the newly produced ITS_LIVE synthesized record of Antarctic Ice Sheet elevation change will
improve understanding of the underlying processes driving the patterns of elevation change, with the hope that
such understanding will aid better projections of ice sheet and sea level change.








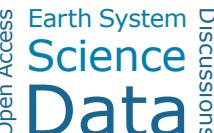

**Competing interests**

The authors declare that they have no conflict of interest.

**Author contributions**

JN contributed to article composition, article figures, article concept, conceptual product design, implementation, and product generation. AS contributed to project initiation, supervision, proofreading, article concept and conceptual product design. FS contributed to via algorithm development, proofreading, git-repository creation and management.

**Financial support**

The authors were supported by the NASA MEaSUREs, and the NASA Cryosphere Science Program (NNH16ZDA001N-ICESAT2), and the Jet Propulsion Laboratory, California Institute of Technology, through an agreement with the National Aeronautics and Space Administration.

**Acknowledgements**

We thank the NASA and the European Space Agency (ESA) for distributing their radar altimetry data. The author would like to thank Sebastian Bjerregaard Simonsen for the discussions and data support during the early part of the study, it was immensely helpful. Further, we would also like to thank Ludwig Schröder for his help with obtaining the Geosat data. © 2021. California Institute of Technology. Government sponsorship acknowledged.



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
