# Peer review of "Elevation Change of the Antarctic Ice Sheet: 1985 to 2020"

_Earth System Science Data, 2021_

## Referee Comment (RC2)

**Review:  Elevation change of the Antarctic Ice Sheet: 1985 to 2020**

Johan Nilsson, Alex S. Gardner, and Fernando S. Paolo

**Summary**
The authors present an integrated satellite altimetry record for the Antarctic ice sheet for the period 1985 through 2020, combining altimetry measurements from seven missions including Geosat, ERS-1, ERS-2, Envisat, ICESat-1, Cryosat-2 and ICESat-2.   The authors develop a series of processing steps to correct for various errors and differences between sensors, including spatial coverage and instrument characteristics.

**General Comments**
The manuscript is generally well written and the authors have described in extensive detail the methods used to generate the dataset.   The suggestions below are mainly related to improving clarity of the manuscript.

**Specific Comments**

1. **Line 16:** Could the authors briefly mention what is novel about the approach?
2. **Figure 1:** The meaning of the concentric circles and mission labels on the map is not entirely clear from the figure.  Please clarify in the text that the circles indicate that there is missing data south of the specified circle for each mission.  The meaning of the basins and basin numbers should be mentioned in the caption.  Labels of different regions of Antarctica would be helpful for interpreting the text later in the manuscript. Also it would be best to include a small colorbar indicating the velocity scale.
3. **Lines 130-131:** Could the authors briefly mention what the 2 m threshold is applied to?
4. **Lines 139-140:** How does this compare to the LRM resolution?
5. **Line 155:**  Please elaborate briefly on how the "relocation method" works.
6. **Lines 156-158:**  Bedmap2 would give more recent estimates of surface slope.   Could this bias estimates from earlier periods and what might be the magnitude of this error?
7. **Lines 160-168:** It would be useful to have a simple schematic diagram summarizing the processing steps, as they are quite extensive and it is easy to lose sight of the bigger picture.
8. **Lines 176-180:** While the authors stated earlier that this was done separately for each mission and mode, it would be helpful to clarify here as well that (1) is done for each mission separately, and to clarify that the purpose of (1) is to establish a different "correction grid" that is suitable for each mission.
9. **Line 183:** By "center date" do the authors mean the center date of the mission?  After establishing the mean topography for ascending and descending orbits separately, do the authors also then remove the mean for

each orbit type within each mission separately?  Please provide some additional details for clarity.

10. **Line 185:** Clarify that the "mathematical model" is a model of the surface topography.  Also clarify that the differing number of coefficients is allowed to vary spatially.

11. **Lines 192-193:** Clarify that this is done for each mission and mode separately.

12. **Line 195:** Clarify that this selection of the best correction occurs when there are multiple centroids per data point.

13. **Line 205:** I don't think this point about a linear temporal trend was not mentioned earlier.  Can the authors mention this earlier in this section and briefly elaborate?

14. **Line 220:** Can the authors briefly explain why variability in the waveform shape can be used to remove these errors?

15. **Line 231:** Are the standard deviations here the standard deviations for the residuals of the Bs, LeW and TeS terms?  If so I think the authors should use "dBs", etc. for the subscript of $\sigma$ to be consistent with "dh".

16. **Line 264-265:** Can the authors provide additional explanation as to which features are related to surface slope and meteorological signals?

17. **Line 288:** Here $n_f$ =2 is specified, but earlier a "four-term" Fourier series is mentioned.  Should $n_f$ = 4?

18. **Lines 292-293:** How does it work for the 2 km grid cell to have a 1 km search radius in some instances – could some data be ignored in this case?

19. **Lines 292-301:** The 10 MAD correction is mentioned twice in this paragraph.  Are these two different steps?  Please clarify.

20. **Line 306:**  Are Envisat and ICESat somehow combined here, or do the authors mean that a coefficient is created for ERS-2 to Envisat and ERS-2 to ICESat.  In general, it is not entirely clear how Envisat and ICESat are combined in the final product.  This should be discussed in further detail.

21. **Line 342:** Are these produced after the first calibration step?  Please clarify.

22. **Line 343:** Again, how has Envisat been grouped with ICESat?

23. **Line 353:** For someone unfamiliar with these regions, it is difficult to know where they are on the map.  Perhaps additional labels can be added to Figure 1, as noted above.

24. **Line 356:**  The authors have mentioned the offsets associated with the initial least squares adjustment, but have not provided any discussion of what happens after the final cross-calibration steps, which could be provided briefly here.

25. **Line 368-369:**  Suggest revising to read "After removal of the long-term trend, the amplitude normalization was computed for each mission, except for ICESat and CryoSat-2, according to:"  so it is clear the formula is for the amplitude noramalization.

26. **Line 372:** It appears that $a$, $\alpha_s$ and $\alpha_c$ are not defined.  Please add definitions for these.

27. **Lines 381-386:** This discussion of the figure would be more appropriate in the main text.
28. **Line 395:** Is this the REMA DEM? Please clarify.
29. **Lines 406-407:** Clarify how the local surface roughness is determined. From the REMA DEM?
30. **Lines 426-428:** The description here is a bit confusing. Which model is added back? Is it a surface extrapolated from ERS1/2 and Envisat?
31. **Lines 441-442:** Can the authors add a brief explanation as to why this procedure was appropriate for the analysis in the paper but not for the data product?
32. **Lines 464-479:** I don't believe Table 1, or the description of the computed errors in Table 1, are mentioned in the text. Additionally, is there a column missing from Table 1 which provides the computed "sensor and mode error" for each sensor? How do these calculated errors fit in to the rest of the error analysis? Please clarify and revise the text appropriately.
33. **Line 496:** Mention here that the validation is discussed further in Section 5.
34. **Line 506:** Make clear that "JPL" is referring to the product described in this paper. Also, I would suggest referring to the authors' product as the "ITS_LIVE" product throughout if the authors think that is reasonable.
35. **Line 541:** I believe the reference to Figure 7c should be changed to "Figure 7e".
36. **Line 542:** The reference to Figures 7d-e should be changed to "7f-g".
37. **Line 544:** "Figure 7b" should be changed to "Figure 7c"
38. **Lines 542-545:** Figure 7b (the comparison between JPL and ATM products) is not mentioned in the text. This should also be mentioned here.
39. **Line 545:** I believe "the synthesis" refers to the JPL/ITS_LIVE product, but this is unclear from the text. Please clarify.
40. **Lines 545-548:** Here Figure 7a can be referenced.
41. **Line 555:** Are the bias and error shown in Fig. 7f-g the bias and error for the JPL product? Please clarify.
42. **Lines 568-573:** Could the authors mention briefly which source data were used for the TUD and CPOM products?
43. **Line 599:** Can the authors note the general location of basin 3?
44. **Line 601:** Change "we find for the 1992-2017…" to "we find the following volume change estimates for the 1992-2017…"
45. **Line 607:** I believe "extrapolation" refers to the hypsometric extrapolation. Please clarify.
46. **Lines 607-608:** The statement that the products agree well but are biased low seems contradictory. Please clarify. Also clarify that "low" refers to lower in magnitude.
47. **Line 611:** Again, clarify that "low" refers to lower in magnitude.
48. **Line 642 (Figure 9):** I believe that the upper-right figure should read "JPL: 1992-2016" rather than "FDM: 1992-2016".
49. **Lines 652-660:** Somewhere in here the authors should reference Figures 10 and 11.

50. **Line 659:** Specify the year rang for the "last decade" and the period for "EAIS has reverted back…".
51. **Lines 667-670:** The trends over the other periods in the EA2 region should be discussed so that the 1985-1989 period can be understood in context.
52. **Lines 682 and 683:** Both Figures 10 and 11 could be referred to here, as it helps to look at these figures when interpreting the text below.
53. **Line 690:** Figure 11 could also be referred to here.
54. **Line 701:** The "earlier timing" is not apparent from the figure, but it is clear that there is a large increase in precipitation during 2006. Please clarify.
55. **Lines 704-706:** Figure 10 could be referred to here.
56. **Lines 717-718:** The strong negative trend for WAIS over the full timeseries should be mentioned here.
57. **Line 732:** Are the authors referring to improvements over previous work? Please clarify.
58. **Line 772:** It is not entirely clear what is done in Figure 12. Is there a processing step additional to what was discussed in the methods section? Please clarify.
59. **Line 791:** What method is being referred to here. Does the correction discussed in the methods section bias the amplitude in the interior of the ice sheet, or is this what would happen if ICESat-2 were to be used?
60. **Line 811:** Should this read "south of 81.5°S" rather than "north"?
61. **Line 815:** What is the bias referred to here relative to?
62. **Line 845:** Could the closer agreement with laser-altimetry validation data be affected by the inclusion of the laser-altimetry data in the development of the ITS_LIVE dataset?
63. **Lines 852-853:** But the authors do mention above this point that there can be errors in the altimetry data in this region. Could this affect the comparison with the firn densification models. Please clarify.
64. **Line 877:** Suggest adding "Our dataset indicates that…" before the beginning of this sentence.

**Technical Corrections**

1. **Line 9:** Change "losses" to "loses".
2. **Line 11:** Change "sea levels rise" to "sea level rise"
3. **Line 53:** Change "inter mission" to "inter-mission"
4. **Line 100:** I believe these are two different flags. If so, then revised to read "quality flags" and "were used". Also, is it "chirp" rather than "chip"?
5. **Line 123:** Change "that uses 532 nm laser" to "that uses a 532 nm laser".
6. **Line 125:** Change "arrange" to "arranged"
7. **Lines 170-190:** To improve readability I would suggest creating a new paragraph for each correction (1,2, and 3), or add italics in the text, e.g. "*Issue (1):* To account for differences in orbital geometry when applying the correction…"
8. **Line 259:** Figure 5 is mentioned before Figure 4. Suggest switching the two.

9. **Line 267 (Figure 3):** Change "CS-2 LRM" and "CS-2 SIN" to "Cryosat-2 LRM" and "Cryosat-2 SarIn" for clarity.
10. **Line 281:** Correct "Bevis etl al."
11. **Line 436:** Suggest revising to "To estimate volume changes at the basin scale (Figure 1)" so that it is clear that "Figure 1" provides the basin outlines and not the volume changes.
12. **Line 450:** I believe Fig. 6 should be referred to here, rather than Fig. 2.
13. **Line 565:** Change "where ICESat-2 shows" to "with ICESat-2 showing".
14. **Line 584:** Change "(as CPOM only provides rates in five-year intervals of all products..." to "(as CPOM only provides rates in five-year intervals) for all products"
15. **Line 599:** Change to read "Larger differences between JPL and CPOM compared to JPL versus TUD..."
16. **Lines 603-604:** Change to read "The regional estimates agree well among products, with the largest discrepancies found in the Antarctic Peninsula."
17. **Line 621:** Change "average surface temperature, 10 m windspeed..." to "average surface temperature, and 10 m windspeed...".
18. **Line 635:** Change "-25 km$^3$ a$^1$" to "-25 km$^3$ a$^{-1}$".
19. **Lines 649-650:** Change "East Antarctic Ice Sheet (...) that experienced" to "East Antarctic Ice Sheet (...), which experienced"
20. **Lines 651:** Change "measure elevation change is" to "measure elevation change of is".
21. **Line 666:** Change "the precipitation event" to "a precipitation event".
22. **Line 672:** Change "panning1985 to 2020" to "spanning 1985 to 2020".
23. **Line 688:** Change "Basin 18, that contains..." to "Basin 18, which contains..."
24. **Line 719:** Change "landfalls" to "landfall".
25. **Line 743:** Change "importance of this correction that can" to "importance of this correction, which can"
26. **Line 756:** Change "that large ice sheet wide changes occur" to "the large ice sheet-wide changes that occur"
27. **Line 769:** Change "long-time separation" to "long time separation".
28. **Line 777:** Change "applied to the data align" to "applied to the data to align".
29. **Line 792:** Change "than than" to "than that"
30. **Line 796:** Change the comma after "maximum orbital coverage" to a semicolon.
31. **Line 827:** Change "contribution significantly" to "contribute significantly".
32. **Line 830:** Change "overall uncertainty estimates" to "overall uncertainty estimate".
33. **Line 831:** Change "ice sheet but rapidly increase closer" to "ice sheet, which rapidly increases closer".
34. **Line 841:** Change "with difference within" to "with differences within"
35. **Line 848:** Change "across mission" to "across missions".
36. **Line 866:** Change "captoolkit - Cryosphere Altimetry Processing Toolkit" to "Cryosphere Altimetry Processing Toolkit (captoolkit)".

---

## Author Comment (AC1)

Response to review from Veit Helm

Firstly, the authors would like to thank the reviewer for taking the time to review this paper and for the kind words!

Here is the point-by-point response to the reviewer's questions in blue text:

Review: Elevation Change of the Antarctic Ice Sheet: 1985 to 2020

Nilsson et. al.

The paper provides a new data set of elevation change of Antarctica based on analysis of satellite altimetry.

The authors put in a massive amount of work to provide this comprehensive dataset. Thanks to the authors.

The authors used for most of the time L2 data from earlier mission for their study. They applied a slope correction (similar to Schröder et.al.) and which is new, included ICESAT(2) data in their study. For CryoSat-2 they applied an independent processing using L1B waveform data.

A new approach to derive elevation change is the decoupling of time-variable and static topography. Then both spatial and temporal patterns of changes in the scattering horizon are estimated and corrected for using waveform parameters and sigma in a separate step Combination of the different mission, which is very challenging was done using a least square approach which also differs from other studies. Another new step is the amplitude normalization to further reduce seasonal amplitudes explained as residuals of not fully corrected radar scattering effects. Finally, an extrapolation method is suggested to fill the pole hole south of 81.5°/88° to provide a complete dataset for Antarctica to be used e.g. for model input. The new data set shows slightly higher accuracy then the TUD and CPOM dataset when compared to ATM or the 2003-2019 ICESat(2) data set of Smith et.al.

To my opinion the paper as well as the data set is of high quality.

The dataset could be accessed through the given link.

I think the data set is significant, unique, useful, and complete and of value for the community and worth to be published.

Language and figures are excellent.

Methods applied are described in detail. Validation as well as comparison to similar datasets are thought-out and well explained.

Results and the development of elevation change of Antarctica over the last 3 decades are well presented.

I have some open question in regard of the uncertainty estimate - see below.

and some minor points:

1.

I was not comfortable with the data handling:

I have the feeling that the chunk dimension of the data in the netcdf file is not applied or is too small or applied to the wrong dimension (I'm not sure). This makes it extremely difficult to read in the data in an acceptable amount of time.

E.g. using ncview takes ages to scroll through the layers. I also used IDL to readin the data and it took very long. Finally, I converted the data myself to be able to check the data quality in a reasonable amount of time. I used a chunk dimension of (x_dim,1,1) when writing my own netcdf file. This improved the access time.

Yes, thank you for that it has been under discussion inside our group as well and we took action to improve the reading time of the product by modifying the current chucking scheme.

2. Error estimate

A very important point is your error estimation. To me it's not clear if you used uncertainties estimated in 4.1 in the integrated error estimation of 4.3. It also not clear what is provided in the product.

Thank you for pointing this out we have made this clearer in the text of what is applied and at what time. Further we have also removed one of the error components (sigma-x) as it was in error. The new total error assigned to each epoch in each time series is now only dependent on the cross-calibration error (sigma-c) and the spatiotemporal variability inside each monthly time interval (sigma-m). Table 1. now reflects the overall quality of the different missions and modes and is mapped directly into monthly uncertainty via the variability. The text has been rewritten to reflect this and that the error varies both by location and time. L497-512

sigma_x I guess are fixed numbers for each mission or mode. Can you provide a table?

Is sigma_c spatially varyiing? I don't get how sigma_m is computed, which variability is used? Please give more details.

Is sigma_m computet for each grid cell?

In table 1 sigma_rand and slope are shown. How is this used in equation 5?

We agree that is somewhat confusing and we have now made clear that this shows the overall mission/mode error, but is not used specifically in the time-series uncertainty. We have re-written a large part of the initial section of 4.1. L497-512

In section 4.3 you estimate sigma_s and sigma_r by comparing to the Smith product. This might make sense for the same time period, but is this valid for the period before 2003?

We totally agree with the reviewer that sigma-s and sigma-r are not appropriate as a stand-alone error for data before the 2003 time period, as it would provide the same error for previous missions as the missions after 2003. We can also see from the crossover-analysis that the relative precision of the different missions decreases as you go back in time. This is the reason that we included the third-term in the error budget to better account for the inherent error for each mission over the different time intervals. Our reasoning is that the systematic bias estimated from the IS1/IS2 validation over the 16-the period provides a baseline. This baseline is then modulated with the inclusion of the random error sources estimated from our analysis. We believe that this provides a conservative estimate of an absolute error. Previous studies have only used relative variability in their estimates and/or errors estimated over short time-spans or regions from airborne data.

In equation 7 sigma_m is mentioned. Is this the same sigma_m as in equation 5 and is this the rmse in your product?

Yes, we apologize that was not real clear and this has now been changed to reflects this better. The change was a part of the re-write of Section 4.1 and you can find specific changes on L508-512 and a change in eq.5. As we also said to the other reviewer there was a small mistake in the equation (double counting one of the error sources) which has now been corrected.

E.g. As a user of your product I'm interested in Pine Island drainage basin for the period 2011 to 2015. How is the procedure to estimate the uncertainty for this given time period for this specific basin and for total Antarctica? Can this be done with the information given in the data product?

This can be done in the following way: Select the basin (ROI) and the time interval from the cube (slice the data in time and mask out areas you are not interested in). Then take the monthly spatial error fields and compute the integrated error for each month. Once, that's has been accomplished compute either the mean or RSS of the errors over the time interval. This will provide the random error (sigma-m) for your ROI over the selected time interval. This can be mapped into an error rate either by dividing by the time interval or by using equation 7. To provide the standard error you can divide the rate-error by the number of un-correlated grid-cells using the correlation length provided in Table 2 (sigma-h^dot). The absolute error can be derived by following equation 7 using the bias (sigma-s), error (sigma-r), area and correlation lengths from Table 2. This follows the approach outlined in this study to provide the tabulated errors in Table 3. Further, if needed sigma-r and sigma-s can be replaced by values from the error model in Figure 7(g,f), which are then integrated over the ROI and substituted into equation 7 using an appropriate correlation length.

It's also not clear if interpolated, extrapolated and observed gid cells are handled in a different way. Furthermore, the uncertainty given for the pole hole looks pretty strange. The whole area is extrapolated but shows very large spatial differences in the RMSE, can you explain why?

The extrapolated grid-cells are handled a bit differently and the predicted errors from the algorithm are multiplied by three to make sure they are not too small and to downweigh them if they are used in time-series as weights. Further, as the only the closest 200 points of the 20 km averaged data are used, it can create a funnel pattern. This has only data from one side of the pole-hole is used in the extrapolation. This can be seen in the error-fields where large errors from the transantarctic mountains are funneled to the pole due to data location. However, this can just be removed by replacing it with a mean-error using the provided pole-mask. It's just our "best" try to provide something consistent and hopefully usable based on actual "local" data. We have revised this section of text in the manuscript to detail this behavior. L456 – 476.

3.

In the text you mention that the bedmap2 elevation model was used. However, the data set itself provide a different elevation model. This is not consistent.

Furthermore, I'm wondering why the old Bedmap2 data set is used instead of using the Tandem-X or REMA DEM's. They are much more accurate and provide reliable data in areas south of 86°.

The bedmap2 DEM was only used for the slope-correction of the older missions as the data used to make the DEM overlap in time with the missions (bedmap2 contains both ERS and ICESat data). For the slope correction we found that a resolution of about 2-3 km provide the best results, thus a high-resolution model was not needed. In general, the only mission that reaches above 86 S that needs a slope correction is CryoSat-2 LRM. The slopes are very gentle in this region and in combination with resolution of the DEM the magnitude of the correction is small. The model provided is of auxiliary use if the user wants to look at time-evolving topography. We have made this clearer in the text. L480-485

4.

Why is it not possible to use Envisat data after the orbit change? Did you try with and without or weren't you able to get good dhdt estimates due to data coverage issues?

Line 130: what exactly is a segmentation filter. How does it work? Please explain in more detail. With the new reference orbit, the tracks are not on the older repeat tracks anymore and thus when removing topography, we found that the noise levels went up due to the need for either a larger search radius and/or it produced an offset in-between the two Envisat orbits. This provided an issue when trying to cross-calibrate the RA2 vs CS2, at least for our study. We are currently working on a way to mitigate this and have made some headway on this and this will be included in the next version of the product.

The segmentation filter is just a difference filter for the IS2 point data to remove data points above a specific threshold (it comes from Smith et al 2020). We compare point "i" with point "i+1 "and if the difference is larger than a threshold its removed/flagged. Its currently set to 2 m which has been derived empirically. We added some text to explain this (L134-135).

5.

To me it's not clear how you handled the different ERS modes. How is the coverage of ICE and OCEAN modes? Do you have data in both modes for each month covering the whole ice sheet or

separate months with ICE or OCEAN or specific areas with one mode for the whole period?

I miss in Fig.3 the ERS1/2 OCEAN mode.

ERS-1/2 are divided into ocean and ice data sets (like in Schroder et al 2019) and treated as different missions and processed independently (topography, scatter correction, bias estimation etc.). However, the ocean mode does not contain that much data so that's why we didn't include any figures as they do not provide that much information. We have updated the text to make the treated clearer. (L314-319)

6.

The combination of Enviat and ICESAT is not fully explained.  At which point you combine these two products? The same for ERS1/2 ocean and ERS1/2 ice.

Do you combine it before the multi-mission cross calibration is applied? Maybe it's worth to show two examples' figures of certain grid cells how such a combination works (e.g., an grid cell with data gaps and without).

All missions or modes (ocean, ice, lrm, sarin etc) are initially adjusted using the least squares approach. The nifty thing with this approach is that at each local-grid cell all data are collected and a mean-offset relative to a target date is estimated for all the different datasets. These offsets are then subtracted providing a cross-calibrated time series (step-1). In the first step independent offsets are estimated for LRM, SIN, E1-OCEAN, E1-ICE, E2-ICE, E2-OCEAN RA2, IS1 and IS2. Once the first cross-calibration has been applied the missions are grouped and a secondary adjustment is performed. This is done to account for more non-linear behavior using the residuals to the model. We group the missions as follows: ERS-1 (Ice+Ocean), ERS-2 (Ice+Ocean), RA2/IS1, CS2 (LRM+SIN) and IS2. Then we compute offsets in-between these grouped using the data where they overlapping in time. Once, the data has been adjusted for the second time we integrated into a consistent time series using a weighted average. We have added some more text to make this clearer and a figure of the cross-calibration is provided in the supplementary material (Figure S1). (L314-319, L332-335)

7.

How much does the normalization of seasonal amplitudes change the trend estimates?

I'm not sure if this kind of normalization should be applied. The point is that you apply correction based on correlations with sigma, LE and TE for each mission. Those correlations make sense and reduce the seasonal amplitude. However, the normalization has no physical explanation. Maybe it's' worth to check and show the seasonal amplitude of the CryoSat L2 product. If the amplitude is similar to your own processed product than the normalization is questionable.

Otherwise, you have an argument that due to your low-level threshold retracking the time varying signal penetration is strongly reduced. Maybe it's also worth to show in an Appendix for each mission the Antarctic wide reduction of seasonal amplitude. I think this can help to understand where the corrections have largest impact and where largest amplitudes are observed and if this is mission specific.

It's also not clear at which point this normalization is applied - before or after the mission cross calibration?

If it is applied before, then you should change the order of the sections in the text.

We knew that using the normalization would be a subject of discussion which is very welcomed. To answerer your first question: No, there is no major effect on the trend as the trend is removed before the correction is applied and then added back. The normalization effect is based on previous work where we did show the difference in seasonal amplitude (Nilsson et al. 2016) between different processing approached for CryoSat-2. This using our own CS2 product compared to the standard ESA L2 product. The results showed that there was clear difference in the magnitude of the seasonal amplitude depending on the choice of retracking technique. Both in that study and in the one presented here we see a much better agreement in amplitude in-between CS2, IS1 and IS2. This holds for both ice sheets when performing the same analysis in Greenland. CS2, IS1 and IS2 are almost perfectly aligned in seasonal amplitude compared to the older mission (RA2, ERS-1 and ERS-2). So, we believe that our justification of applying the correction is valid for the older missions. This as it is obvious that the scattering correction is not fully capable of removing all the artificial signals. Further, as an independent comparison we did compare the amplitude against the RACO FDM product and found a remarkable agreement. This further gave us confidence for the justification of the correction. We have added tables in the supplementary material (Table S1/S2), where we compare the amplitude of the RACMO FDM product to the amplitude estimated form our product over different time periods (missions and total) to help illustrate this in more detail. The normalization correction is applied after the calibration step. Data quality-wise it made little difference if it was applied before or after.

8.

Line 450: Here the reference to the figure 2 is not correct. It should be Figure 6.

Line 517: You mention a correlation length of 100km, however Table 2 list different values. Which one was used?

We used 100 km for the correlation length. (L575 and L462)

9.

Line 608: Do you have any idea why your product is not closer to LA in WAIS? For EAIS and AP they are and this is what I would suppose, as you also used ICESAT in your approach?

This is in my view a classic example of the difficulty of radar to measure in high slope areas. The radar can't capture the same amount of signal as laser, as also seen by other products. IS1, due to its lower temporal and spatial sampling, is kind of over-shadowed by RA2 here. Maybe in the future version of the product we might be able to better account for that?

---

## Author Comment (AC2)

**Response to Reviewer - 2**

We would like to thank the reviewer for taking the time to review and comment on this study which helped to improve the manuscript.

1. Line 16: Could the authors briefly mention what is novel about the approach?
This has been added to the text. L16-17

2. Figure 1: The meaning of the concentric circles and mission labels on the map is not entirely clear from the figure. Please clarify in the text that the circles indicate that there is missing data south of the specified circle for each mission. The meaning of the basins and basin numbers should be mentioned in the caption. Labels of different regions of Antarctica would be helpful for interpreting the text later in the manuscript. Also, it would be best to include a small colorbar indicating the velocity scale.
Changes have been made to the figure and the caption has been expanded.

3. Lines 130-131: Could the authors briefly mention what the 2 m threshold is applied to?
We have added more detailed explanation regarding this. (L134-135)

4. Lines 139-140: How does this compare to the LRM resolution?
LRM resolution is on the order of 1.5 km (radius) while SARIn has a 350 x 1500 m resolution. Expanded text to better explain this. (L145-146)

5. Line 155: Please elaborate briefly on how the "relocation method" works.
We have elaborated more on the relocation method in the text. (L175-177)

6. Lines 156-158: Bedmap2 would give more recent estimates of surface slope. Could this bias estimates from earlier periods and what might be the magnitude of this error?
Bedmap-2 was selected as it mostly contains data from ERS-1 and ICESat and thus provide a DEM that coincides in time with data altimetry we are trying to correct. We do not believe that it will cause any bias as both ERS-1, ERS-2, Envisat and LRM are corrected with the same dataset. One has to also remember that we do not need to correct any of the SARIn or laser altimetry data for the slope induced error. The only new dataset that needs to be corrected is the LRM data, but the data is located in the interior of the ice sheet which is very stable and see little change. Hence, that the DEM is older does not introduce any major issue in our view.

7. Lines 160-168: It would be useful to have a simple schematic diagram summarizing the processing steps, as they are quite extensive and it is easy to lose sight of the bigger picture.

We have in the "Methods" section added a paragraph summarizing the different steps and in what order they are applied. (L159-167)

8. Lines 176-180: While the authors stated earlier that this was done separately for each mission and mode, it would be helpful to clarify here as well that (1) is done for each mission separately, and to clarify that the purpose of (1) is to establish a different "correction grid" that is suitable for each mission.

We have rewritten and added some text to better described this. (L191-214)

9. Line 183: By "center date" do the authors mean the center date of the mission? After establishing the mean topography for ascending and descending orbits separately, do the authors also then remove the mean for each orbit type within each mission separately? Please provide some additional details for clarity.

The center date in this case is the mean time of the mission and is used to center the "Asc" and "Des" time series to the elevation [h(t0)]. Yes, we remove the spatial contribution of the model from both the A and D orbits (not the linear temporal term). The inclusion of the linear term temporal term, centered a specific date, will aligned the data to the same residual elevation (intercept is zero for both at reference time). We have added some clarification in the text for this. (L191-214).

10. Line 185: Clarify that the "mathematical model" is a model of the surface topography. Also clarify that the differing number of coefficients is allowed to vary spatially.

We have added text to better describe this. (L207-208)

11. Lines 192-193: Clarify that this is done for each mission and mode separately.

This has been done. (L198, L203-204)

12. Line 195: Clarify that this selection of the best correction occurs when there are multiple centroids per data point.

Not entirely true as stated. If their already exists a solution in the output vector at a specific location that solution is only over-written if the new solution has a lower RMS. Added some more clarification about this. (L219-222)

13. Line 205: I don't think this point about a linear temporal trend was not mentioned earlier. Can the authors mention this earlier in this section and briefly elaborate?

This has been added. (L193, L205, L212-213)

14. Line 220: Can the authors briefly explain why variability in the waveform shape can be used to remove these errors?
This has been added. (L245-247)

15. Line 231: Are the standard deviations here the standard deviations for the residuals of the Bs, LeW and TeS terms? If so I think the authors should use "dBs", etc. for the subscript of s to be consistent with "dh".
These are standard deviation from the variables themselves (not the residuals), so we did not provide any change in the manuscript related to this.

16. Line 264-265: Can the authors provide additional explanation as to which features are related to surface slope and meteorological signals?
We added some extra text to describe this. (L290-293)

17. Line 288: Here $n_f=2$ is specified, but earlier a "four-term" Fourier series is mentioned. Should $n_f = 4$?
Good catch! We have changed it to four!

18. Lines 292-293: How does it work for the 2 km grid cell to have a 1 km search radius in some instances – could some data be ignored in this case?
In 99% of the time a 1 km search radius is too small and does not provide enough data to pass the 70% fill criteria. In reality we needed at least a search radius of 2-3 km to pass the 70% criteria. To avoid data being "lost" the algorithm increases the radius until 10 km is achieved for any node. If 10 km is reached and the time series is not filled it uses the available data. Some extra text has been added to better describe the process. (L326-327)

19. Lines 292-301: The 10 MAD correction is mentioned twice in this paragraph. Are these two different steps? Please clarify.
Yes, these are two different steps: The first step is a 1-year windowing filter and the second one is filtering based on model residuals. We have modified the text slightly to better explain this. (L333-335)

20. Line 306: Are Envisat and ICESat somehow combined here, or do the authors mean that a coefficient is created for ERS-2 to Envisat and ERS-2 to ICESat. In general, it is not entirely clear how Envisat and ICESat are combined in the final product. This should be discussed in further detail.
This was asked by the other reviewer as well and we have made it much more clear in the text how these offsets are generated and combined. We hope that it will clear up any confusion. (L318-335)

21. Line 342: Are these produced after the first calibration step? Please clarify.
See (20)

22. Line 343: Again, how has Envisat been grouped with ICESat?
See (20)

23. Line 353: For someone unfamiliar with these regions, it is difficult to know where they are on the map. Perhaps additional labels can be added to Figure 1, as noted above.
We have added some notation on Figure-1 that we hope will help. (L70-74)

24. Line 356: The authors have mentioned the offsets associated with the initial least squares adjustment, but have not provided any discussion of what happens after the final cross-calibration steps, which could be provided briefly here.
To reduce confusion, we added the step-by-step description in the beginning of Section 3 and expanded the text in the end of Section 3.2.3. We hope that fixes the issue. (L156-167)

25. Line 368-369: Suggest revising to read "After removal of the long-term trend, the amplitude normalization was computed for each mission, except for ICESat and CryoSat-2, according to:" so it is clear the formula is for the amplitude normalization.
This has been done

26. Line 372: It appears that a, as and ac are not defined. Please add definitions for these.
This has been done. (L414)

27. Lines 381-386: This discussion of the figure would be more appropriate in the main text.
We have moved some of this to the normalization section. (L419-422)

28. Line 395: Is this the REMA DEM? Please clarify.
No this is not REMA its bedmap2 we have made that clear in the text. (L437 and L480)

29. Lines 406-407: Clarify how the local surface roughness is determined. From the REMA DEM?
It's estimated from bedmap2, we have added this to the text. (L480)

30. Lines 426-428: The description here is a bit confusing. Which model is added back? Is it a surface extrapolated from ERS1/2 and Envisat?
We have re-written some of the text in this section to make this much clearer. (L456-476)

31. Lines 441-442: Can the authors add a brief explanation as to why this procedure was appropriate for the analysis in the paper but not for the data product?
A better explanation was added to the text. (L493-494)

32. Lines 464-479: I don't believe Table 1, or the description of the computed errors in Table 1, are mentioned in the text. Additionally, is there a column missing from Table 1 which provides the computed "sensor and mode error" for each sensor? How do these calculated errors fit in to the rest of the error analysis? Please clarify and revise the text appropriately.

There was an issue with the table numbering which is now fixed. You will find that Table-1 is now referenced several times, as is Table 2 and Table 3. The other reviewer asked the same question: Table 1 is now used only to provide the user a view of the overall quality of the missions, but are not used in the error calculation. Here, sigma-m represent or contains this error. This can be seen in the time-series by looking the magnitude of the error bars, where the older missions have larger errors. We have revised a large section of the text here. (L497-512)

33. Line 496: Mention here that the validation is discussed further in Section 5.

Done

34. Line 506: Make clear that "JPL" is referring to the product described in this paper. Also, I would suggest referring to the authors' product as the "ITS_LIVE" product throughout if the authors think that is reasonable.

Clarification has been added. We have chosen to retain the JPL name for now. This to allow it to be more processing center specific when comparing.

35. Line 541: I believe the reference to Figure 7c should be changed to "Figure 7e".

Thanks for catching this! It has been changed

36. Line 542: The reference to Figures 7d-e should be changed to "7f-g".

Thanks for catching this! It has been changed

37. Line 544: "Figure 7b" should be changed to "Figure 7c"

Thanks for catching this! It has been changed

38. Lines 542-545: Figure 7b (the comparison between JPL and ATM products) is not mentioned in the text. This should also be mentioned here.

This has now been added and discussed more in the text. (L603-605)

39. Line 545: I believe "the synthesis" refers to the JPL/ITS_LIVE product, but this is unclear from the text. Please clarify.

Changed this to product

40. Lines 545-548: Here Figure 7a can be referenced.

Done

41. Line 555: Are the bias and error shown in Fig. 7f-g the bias and error for the JPL product? Please clarify.

We have changed the text to make that clear in the caption. (L607-614)

42. Lines 568-573: Could the authors mention briefly which source data were used for the TUD and CPOM products?
This has been added to the text

43. Line 599: Can the authors note the general location of basin 3?
We have added the general location in the text

44. Line 601: Change "we find for the 1992-2017…" to "we find the following volume change estimates for the 1992-2017…"
We changed it accordingly in the text.

45. Line 607: I believe "extrapolation" refers to the hypsometric extrapolation. Please clarify.
This has been clarified in the text. (L668-669)

46. Lines 607-608: The statement that the products agree well but are biased low seems contradictory. Please clarify. Also clarify that "low" refers to lower in magnitude.
We have changed the wording of the sentence accordingly. (L672-673)

47. Line 611: Again, clarify that "low" refers to lower in magnitude.
This has been added

48. Line 642 (Figure 9): I believe that the upper-right figure should read "JPL: 1992-2016" rather than "FDM: 1992-2016".
Thanks for catching this! We have updated the figure.

49. Lines 652-660: Somewhere in here the authors should reference Figures 10 and 11.
We have added mentions to different figures here

50. Line 659: Specify the year rang for the "last decade" and the period for "EAIS has reverted back…".
Range has been added

51. Lines 667-670: The trends over the other periods in the EA2 region should be discussed so that the 1985-1989 period can be understood in context.
We have added discussion about other trends in EA2. (L730-740)

52. Lines 682 and 683: Both Figures 10 and 11 could be referred to here, as it helps to look at these figures when interpreting the text below.
Added to manuscript

53. Line 690: Figure 11 could also be referred to here.
Done

54. Line 701: The "earlier timing" is not apparent from the figure, but it is clear
that there is a large increase in precipitation during 2006. Please clarify.
We have changed the text slightly and removed "timing"

55. Lines 704-706: Figure 10 could be referred to here.
Added

56. Lines 717-718: The strong negative trend for WAIS over the full timeseries
should be mentioned here.
We have added text about the long-term negative trend in WAIS. (L797-798)

57. Line 732: Are the authors referring to improvements over previous work?
Please clarify.
Yes, as this is the first study that we know of that separates the inversion in this way. We have
added some references. (L813)

58. Line 772: It is not entirely clear what is done in Figure 12. Is there a
processing step additional to what was discussed in the methods section?
Please clarify.
Yes, it is an extra post processing step as the locally estimated correction might not always hold
for larger regions. Hence, when a regional time series is generated, there might be need to re-
align the Geosat data with the other part of the record. This is what was done in Figure 12 and it
is mentioned in the discussion. We have expanded the text in the figure caption to make it
clearer. (L857-859)

59. Line 791: What method is being referred to here. Does the correction
discussed in the methods section bias the amplitude in the interior of the ice
sheet, or is this what would happen if ICESat-2 were to be used?
We removed this paragraph as it was not really needed.

60. Line 811: Should this read "south of 81.5  S" rather than "north"?
Yes, it should be South. Thanks for catching that.

61. Line 815: What is the bias referred here relative to?
Here, the bias is reference in the form of direction. It should be positive but from the other
products they are negative. We changed the text to make this clearer. (L896)

62. Line 845: Could the closer agreement with laser-altimetry validation data be affected by the inclusion of the laser-altimetry data in the development of the ITS_LIVE dataset?
Yes, to a degree but in general the effect is small in our view. The rational being that in areas of rapid change all three products agree well, but contain either little or no laser altimetry data (TUD and CPOM respectively). So, the improvement is more related to the new methodology and use of algorithms. Unfortunately, the ICESat data included in our and the TUD product is overshadowed by the Envisat data, due to it much higher spatial and temporal sampling. It does improve the solution, but due to the sheer number of Envisat points it has less impacts on the validation statistics. The sampling difference of Envisat and ICESat can be seen here: https://www.science.org/action/downloadSupplement?doi=10.1126%2Fscience.aaa5727&file=wouters-sm.pdf in Figure S9.

63. Lines 852-853: But the authors do mention above this point that there can be errors in the altimetry data in this region. Could this affect the comparison with the firn densification models. Please clarify.
Here we were referring to the Geosat data over the Totten region and not all the data. The 1992-2020 data are of good quality. We have added some wording to explain what time period we are referring to. (L935)

64. Line 877: Suggest adding "Our dataset indicates that…" before the beginning of this sentence.
Thank you! It has been added.

Technical Corrections
1. Line 9: Change "losses" to "loses".
Fixed

2. Line 11: Change "sea levels rise" to "sea level rise"
Fixed

3. Line 53: Change "inter mission" to "inter-mission"
Fixed

4. Line 100: I believe these are two different flags. If so, then revised to read "quality flags" and "were used". Also, is it "chirp" rather than "chip"?
Fixed

5. Line 123: Change "that uses 532 nm laser" to "that uses a 532 nm laser".
Fixed

6. Line 125: Change "arrange" to "arranged"
Fixed

7. Lines 170-190: To improve readability I would suggest creating a new paragraph for each correction (1,2, and 3), or add italics in the text, e.g. "Issue

(1): To account for differences in orbital geometry when applying the correction…"
We have added italic font to this paragraph to make it easier to read.

8. Line 259: Figure 5 is mentioned before Figure 4. Suggest switching the two.
We have changed the text to point to Section 3.2.4 as we have moved a lot of the justification from Figure 5 to this section.

9. Line 267 (Figure 3): Change "CS-2 LRM" and "CS-2 SIN" to "Cryosat-2 LRM" and "Cryosat-2 SarIn" for clarity.
We have changed the figure accordingly

10. Line 281: Correct "Bevis etl al."
Corrected

11. Line 436: Suggest revising to "To estimate volume changes at the basin scale (Figure 1)" so that it is clear that "Figure 1" provides the basin outlines and not the volume changes.
Changed

12. Line 450: I believe Fig. 6 should be referred to here, rather than Fig. 2.
Fixed and rewritten

13. Line 565: Change "where ICESat-2 shows" to "with ICESat-2 showing".
Changed

14. Line 584: Change "(as CPOM only provides rates in five-year intervals of all products…" to "(as CPOM only provides rates in five-year intervals) for all products"
Changed

15. Line 599: Change to read "Larger differences between JPL and CPOM compared to JPL versus TUD…"
Changed

16. Lines 603-604: Change to read "The regional estimates agree well among products, with the largest discrepancies found in the Antarctic Peninsula."
Changed

17. Line 621: Change "average surface temperature, 10 m windspeed…" to "average surface temperature, and 10 m windspeed…".
Changed

18. Line 635: Change "-25 km$_3$ a$_1$" to "-25 km$_3$ a$_{-1}$".
Changed

19. Lines 649-650: Change "East Antarctic Ice Sheet (…) that experienced" to "East Antarctic Ice Sheet (…), which experienced"
Changed

20. Lines 651: Change "measure elevation change is" to "measure elevation change of is".
Changed

21. Line 666: Change "the precipitation event" to "a precipitation event".
Changed

22. Line 672: Change "panning1985 to 2020" to "spanning 1985 to 2020".
Changed

23. Line 688: Change "Basin 18, that contains…" to "Basin 18, which contains…"
Changed

24. Line 719: Change "landfalls" to "landfall".
Changed

25. Line 743: Change "importance of this correction that can" to "importance of this correction, which can"
Changed

26. Line 756: Change "that large ice sheet wide changes occur" to "the large ice sheet-wide changes that occur"
Changed

27. Line 769: Change "long-time separation" to "long time separation".
Changed

28. Line 777: Change "applied to the data align" to "applied to the data to align".
Fixed and rewritten

29. Line 792: Change "than than" to "than that"
Fixed and rewritten

30. Line 796: Change the comma after "maximum orbital coverage" to a semicolon.
Changed

31. Line 827: Change "contribution significantly" to "contribute significantly".
Changed

32. Line 830: Change "overall uncertainty estimates" to "overall uncertainty estimate".

Changed

33. Line 831: Change "ice sheet but rapidly increase closer" to "ice sheet, which rapidly increases closer".
Changed

34. Line 841: Change "with difference within" to "with differences within"
Changed

35. Line 848: Change "across mission" to "across missions".
Changed

36. Line 866: Change "captoolkit - Cryosphere Altimetry Processing Toolkit" to "Cryosphere Altimetry Processing Toolkit (captoolkit)".
Changed

---

## Author Response (AR2)

**Response to review from Veit Helm**

The response to the reviewer from the author is show in blue text.

**Suggestions for revision or reasons for rejection (will be published if the paper is accepted for final publication)**

We thank the authors of their detailed response to the 1st reviewer round.
All of the points raised in the 1st round were answered and the manuscript updated with more details of the uncertainty estimate and clarification in the text.

I recognized that the data set was not updated. So, my criticism of the very slow access rate of the data is still valid. The authors responded to take care of this problem, but I could not check if any improvement was made.

We apologies for this issue. There seems to have a been a problem with the dissemination of the product downstream from me/us. NSIDC have now updated the DOI so the new version with improved chunking is now available on the ITS_LIVE website. We have provided the link in the response here below:

http://its-live-data.s3.amazonaws.com/height_change/Antarctica/Grounded/ANT_G1920V01_GroundedIceHeight.nc

The authors also mentioned in their response that the uncertainty sigma_m was changed. This means that RMSE in the netcdf file which reflects (sigma_m in the text) was changed. Is this correct? I think it's worth to add to the netdf file in the long name of RMSE a note that RMSE equals sigma_m.

No, this was an oversight in the actual manuscript itself at the time of writing (related to an old version of the dataset). The product "rmse" did not change and is the same. Further, we have updated the description for the "rmse" variable to point to the sigma_m variable in the paper.

equation 7: Please check. It seems that there is a: above h

We have fixed this

Table 2:
please add in
column2: Bias equals sigma_s^2 and
colums 3: Error equals sigma_r^2/n
at least this is what I understood.

We added this to the caption of the table

Table3:
I don't see that this was updated. When you corrected something in your error budget shouldn't this be reflected in Table 3 as well?

The errors in the product have stayed the same so there is no change in Table-3. Table-1 was not used in the generation of the product errors it was only added for educational purposes to show the overall noise levels of the different missions.

Fig2: Could you please state in the figure caption if the curves were derived at a single grid point or are averaged over the lake area.

We have added the wording "area integrated" to ensure that the reader understand that the depicted time series are for the entire lake.

Suggestion:
One or two components of your area integrated error estimation is based on the difference between JPL and smith elevation rates (2003 - 2019). Why not add this difference field to the netcdf file. This would allow the user to apply your error budget procedure to any region and time period of their choice.

Thank you for the suggestion but we have decided to keep the product as is to keep the size down. However, this might be something to add on in future versions. The IS1/IS2 product can downloaded from Smith et al. (2020) and can be differenced with rates generated from the cube with very little effort. We have expanded the supplement to include this information.

---

## Author Response (AR3)

**Response to editor**

*Editor's comments in black and authors response in blue.*

Line 579, Table 2 caption: double check the statement: "The bias (mean – $\sigma s$ ) and error (standard deviation - $\sigma r$ )", which seems to indicate that bias=(mean–$\sigma s$) and error=(standard deviation-$\sigma r$ ). Please double check to make sure this is correct and it is what you have intended or to make any necessary changes.

We have changed this to read: "bias = (mean: $\sigma s$) and error = (standard deviation: $\sigma r$ )" to avoid confusion on Line 567, Table 2.

For the next revision, I kindly ask you to remove the Copernicus logo from page 41.

We have removed the logo as requested